# Write cycling endurance exceeding $10^{10}$ in sub-50 nm ferroelectric AlScN

Hyunmin Cho, Yubo Wang, Chloe Leblanc, Yinuo Zhang, Yunfei He ⬡ , Zirun Han, Xiaolei Tong, Vidhu D. Bulumulla, Jonathan Tan, Roy H. Olsson III ⬡ ✉ & Deep Jariwala ⬡ ✉

Wurtzite ferroelectrics, particularly aluminum scandium nitride (AlScN), have emerged as a promising material platform for non-volatile memories, offering high polarization values exceeding 100 $\mu C/cm^2$. However, their high coercive fields (>3 MV/cm) have limited cycling endurance to ~$10^7$ cycles in previous reports. Here, we demonstrate unprecedented control of polarization switching in AlScN, achieving write cycling endurance exceeding $10^{10}$ cycles—a thousand-fold improvement over previous wurtzite ferroelectric benchmarks. Through precise voltage modulation in 45 nm-thick $Al_{0.64}Sc_{0.36}N$ capacitors, we show that while complete polarization reversal (2$P_r \approx$ 200 $\mu C/cm^2$) sustains ~$10^8$ cycles, partial switching extends endurance beyond $10^{10}$ cycles while maintaining a substantial polarization (>30 $\mu C/cm^2$ for 2$P_r$). This exceptional endurance, combined with breakdown fields approaching 10 MV/cm in optimized 10 $\mu$m diameter devices, represents the highest reported values for any wurtzite ferroelectric. Our findings establish a new paradigm for reliability in nitride ferroelectrics, demonstrating that controlled partial polarization and size scaling enables both high endurance and energy-efficient operation.

The exponential growth in data processing demands has intensified the search for energy-efficient, high-endurance, non-volatile memories. While conventional memory technologies face fundamental scaling and endurance limitations[1,2], ferroelectric materials offer a promising solution through their non-volatile polarization switching and CMOS compatibility[3,4]. However, achieving both high endurance (>$10^{10}$ cycles) and reliable operation remains a critical challenge for the practical implementation of non-volatile memory technologies[5].

The discovery of ferroelectricity in wurtzite-structured aluminum scandium nitride (AlScN) in 2019 introduced a fundamentally new class of polar materials[6]. Unlike conventional oxide ferroelectrics which rely on oxygen vacancy ordering (HfO$_2$ based ferroelectrics) or B-site cation displacement (PZT), wurtzite ferroelectrics achieve polarization through intrinsic ionic charge separation in a non-centrosymmetric lattice[7,8]. This distinct mechanism enables remarkably high remnant polarization exceeding 100 $\mu C/cm^2$ in AlScN-significantly higher than the typical range of 10–40 $\mu C/cm^2$ observed

in HfO$_2$-based ferroelectrics[3,8]. Moreover, AlScN can be deposited at back-end-of-line (BEOL)-compatible temperatures below 400 °C while maintaining reliable ferroelectric switching in films as thin as 5 nm[9]. These properties, combined with demonstrated operation at temperatures up to 600 °C[10], position wurtzite ferroelectrics as promising candidates for next-generation non-volatile memory applications.

The implementation of wurtzite ferroelectrics in practical memory devices faces two key challenges. First, previously reported AlScN devices show limited endurance, typically failing before $10^7$ switching cycles[11,12]. Second, the high coercive fields necessary for complete polarization switching (>3 MV/cm) lead to significant power consumption and accelerated device degradation[7]. These limitations are particularly pronounced in scaled devices, where local defects can dominate switching behavior and lead to premature breakdown[13].

In this work, we demonstrate a breakthrough in AlScN ferroelectric device reliability through controlled partial polarization switching and systematic size scaling. Using precisely modulated

Department of Electrical and Systems Engineering, University of Pennsylvania, Philadelphia, PA, USA. ✉e-mail: rolsson@seas.upenn.edu; dmj@seas.upenn.edu

voltage pulses in 45 nm-thick $Al_{0.64}Sc_{0.36}N$ capacitors, we achieve write cycling endurance exceeding $10^{10}$ cycles while maintaining substantial remnant polarization above $30\,\mu C/cm^2$. This represents a thousand-fold improvement over previous benchmarks for wurtzite ferroelectrics[11,12]. By reducing device dimensions to 10 μm diameter, we achieve breakdown fields approaching 10 MV/cm, establishing new performance standards for nitride ferroelectrics. Our findings reveal that controlled partial polarization, combined with optimal device scaling, provides a pathway to simultaneously achieve high endurance and energy efficiency in ferroelectric memories.

## Results and discussion

The schematic in Fig. 1a illustrates an $Al_{0.64}Sc_{0.36}N$ capacitor specifically designed to minimize stress on the $Al_{0.64}Sc_{0.36}N$ layer during measurements. This reduction in stress was achieved to avoid the pressure of the probe tip with the via contact. The device features an Al (50 nm) bottom electrode, an $Al_{0.64}Sc_{0.36}N$ (45 nm) layer, and an Al (50 nm) top electrode, all deposited in-situ to ensure minimal interface defects. We used in-situ Al as the top electrode to ensure a clean and stable interface. This immediate capping after AlScN growth prevents surface oxidation and defect formation. As a result, the Al contact provides the most reliable and reproducible device performance. However, the choice of Al was not based on a systematic comparison with other metals. Future studies with alternative electrodes will be essential to fully clarify their impact on AlScN endurance. Additionally, Pd/Ti/Cr (250 nm) was deposited as the via contact material, and $SiO_2$ (200 nm) was utilized below the contact pad to sustain the capacitor structure. This architecture isolates the top and bottom contact pads with a via structure, effectively reducing unwanted capacitance and leakage currents. During operation, voltage was applied to the bottom electrode, while the top electrode was grounded, allowing stable current measurement from the top contact. Furthermore, this design minimizes current detours or breakdown paths between the top and bottom electrode, ensuring reliable operation under consistent measurement conditions. Additional information about the via structure can be found in Supplementary Note 1.

Figure 1b and c shows optical microscope images from the top view. Various top electrode sizes were prepared simultaneously on the same sample to minimize sample fabrication variations. Additionally, the bottom contact pad was positioned consistently for each set of capacitors to ensure reliable measurement results. With this well-prepared device, the current density–voltage ($J$–$V$) characteristics were measured to evaluate the electrical behavior as shown in Fig. 1d. The measurements were conducted with a standard bipolar voltage pulse with a linear ramp-up and ramp-down sequence, configured as a 10 kHz triangular pulse. To ensure consistency, capacitors with varying diameters were tested. The results demonstrated high reliability, showing minimal variation regardless of diameter. An increase in leakage current was observed for positive voltages as the applied voltage approached higher values. We suggest that the asymmetric leakage current originates from polarization-dependent leakage[4,14,15] combined with space-charge-limited current[16]. The N-polar imprint field[17] enhances carrier injection under positive bias while suppressing it under negative bias. Nevertheless, further systematic investigations are necessary to substantiate this interpretation. This behavior led to an overestimation of the $2P_r$, making accurate measurements infeasible for positive applied switching voltages. Therefore, the analysis focused on the data from the negative voltage switching, where leakage effects were minimized[14]. This approach enabled a reliable determination of the device's polarization and switching characteristics. These $J$–$V$ characteristics are similar to our previous reports[4,10]. Additionally, we conducted quasi direct current-voltage (DC-IV) measurements with various diameters, as shown in Supplementary Fig. S1.

Furthermore, in Supplementary Fig. S2, X-Ray diffraction analysis verified the crystallinity of the AlScN, showing a strong (0002) peak

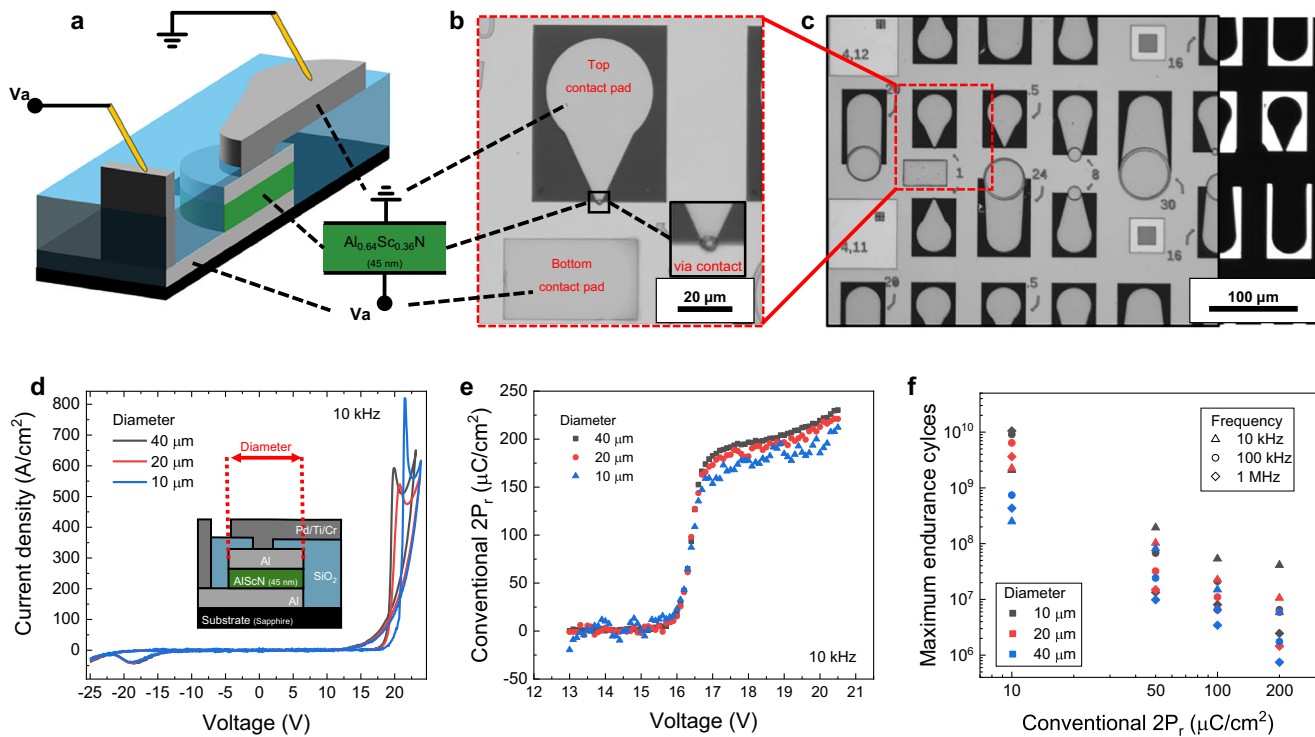

**Fig. 1 | Schematic of AlScN capacitor and its electrical characteristics.**
**a** Schematic illustration of the $Al_{0.64}Sc_{0.36}N$ ferroelectric capacitor with a via-contact structure. **b**, **c** Optical microscope images of the ferroelectric capacitor. The number on the image indicates the radius of the top electrode. Most of the image in **c** was acquired under top illumination, while a small region on the right was imaged under bottom illumination. **d** $J$–$V$ hysteresis loops, **e** ferroelectric polarization response to applied voltage pulses, and **f** maximum endurance cycles vs. conventional $2P_r$ for various diameters (10, 20, and 40 μm).

with no secondary reflections, confirming high-quality c-axis–oriented growth suitable for device fabrication.

To quantitatively analyze $2P_r$, Positive-Up-Negative-Down (PUND) pulsed measurements were performed at a frequency of 10 kHz, using a square pulse width of 50 μs. This methodology is highly effective for isolating the ferroelectric switching current response by minimizing the impact of leakage currents[18]. Consequently, it provides a precise measurement of the intrinsic ferroelectric properties. In this study, the $2P_r$ extracted using the PUND method is defined as the 'conventional $2P_r$'.

Figure 1e shows the relationship between conventional $2P_r$ and applied voltage. This test was conducted across capacitors with varying top electrode diameters. Switching of ferroelectric dipoles were initiated at a voltage pulse of 16 V, with conventional $2P_r$ rapidly approaching a saturation value of ~200 μC/cm². Within the voltage range where conventional $2P_r$ transitions from onset to saturation, partial polarization was observed. This effect is due to incomplete domain switching within the applied voltage range[19–22]. Gradual alignment of ferroelectric domains during this process provides key insights into partial polarization of AlScN. Therefore, this behavior reflects the dynamic switching characteristics of the material. Notably, the partial polarization remained stable across varying top electrode diameters, demonstrating high reproducibility and enabling direct comparison of results independent of electrode size. Supplementary Figs. S3 and S4 present the corresponding time-dependent current response. Additionally, Supplementary Note 2 specifically focuses on PUND results to explain the partial polarization switching, providing further insights into this phenomenon. In addition, Supplementary Note 3 presents piezoresponse force microscopy measurements, which confirm that partial polarization switching indeed occurred in the AlScN.

Partial polarization has emerged as a promising mechanism for advancing multistate memory and neuromorphic devices[4,23–26]. Despite its recognized potential, achieving stable and controllable intermediate polarization states remains a critical challenge. Notably, AlScN presents an exceptional ability to sustain stable partial polarization due to its distinct ferroelectric mechanism[21]. Building on these characteristics, this study explores an innovative approach to enhance endurance cycles by leveraging the intrinsic ferroelectric behavior of AlScN, thereby pushing the boundaries of device performance and reliability.

To further investigate this behavior, additional measurements were performed on a 40 μm capacitor at a different frequency as shown in Supplementary Fig. S3. As the frequency increases, an increase in operation voltage was observed. The increase of operation voltage refers to the applied voltage necessary to obtain the same conventional $2P_r$. Additionally, a difference in the slope of conventional $2P_r$ from onset to saturation was detected, as shown in Supplementary Fig. S3. This indicates a distinct frequency-dependent modulation of domain switching. The observation of increased coercive voltage ($V_C$) is consistent with reports in previous literature[6,9,27,28]. This suggests that the switching kinetics are influenced by the interplay between external field dynamics and the intrinsic material response. This interplay highlights the importance of further exploring frequency-dependent polarization behavior. Understanding these effects could provide deeper insights into domain switching mechanisms.

Figure 1f shows the relationship between conventional $2P_r$ and maximum endurance cycles in AlScN capacitors. To comprehensively investigate this relationship, we conducted additional tests by varying both the frequency of the fatigue voltage pulses and the diameter of the capacitors' top electrode. In each case, at least two to five devices were measured, and similar results were consistently observed as shown in Supplementary Fig. S5. Here, we must pay attention on the x-axis to conventional $2P_r$ values lower than 200 μC/cm², such as 10,

50, and 100 μC/cm². These are the results of partial polarization because of applying a switching voltage lower than $V_C$. These values represent controlled partial polarization switching sustained throughout the endurance measurement, as further detailed in the following paragraph. However, such partial polarization values cannot be accurately extracted using the conventional PUND method which is only suitable for full polarization measurements. Because, based on the explanation provided in Supplementary Note 4, the conventional $2P_r$ values (10, 50, and 100) are always lower than the actual partially switched $2P_r$. However, conventional $2P_r$ values are used on the x-axis for consistent comparison. In addition, we define the actual amount of partially or fully switched polarization as 'intrinsic $2P_r$' in this context.

To test the endurance of partially or fully switched AlScN, each fatigue iteration consists of one positive and one negative pulse with identical time of pulse duration. A detailed pulse train configuration is depicted in Supplementary Fig. S6. During the test, the voltage pulse amplitudes are carefully adjusted to keep the measured conventional $2P_r$ close to a specific $2P_r$ value. Here, we define the specific $2P_r$ targeted in the program as the 'preset $2P_r$'. Therefore, the x-axis in Fig. 1f represents both the conventional $2P_r$ and the preset $2P_r$. This equivalence is ensured by a well-developed program that maintains consistency between the conventional (measured) and preset (programmed) $2P_r$ values with minimal deviation.

A pivotal observation from these experiments is that a reduced conventional $2P_r$ markedly enhances the endurance of the capacitors. This inverse relationship shows that as the conventional $2P_r$ increases, the maximum endurance cycles diminish accordingly. Remarkably, capacitors subjected to a conventional $2P_r$ of 10 μC/cm² exhibit endurance exceeding 10 billion cycles, a performance that significantly outpaces prior reports. Furthermore, reducing the top electrode diameter contributes to a noticeable improvement in endurance. Detailed explanations are discussed later in this paper. Further, it is also noteworthy that even for full conventional $2P_r$ switching equaling ~200 μC/cm², the 10 μm diameter capacitors last up to $10^8$ cycles at a 10 kHz frequency, again surpassing any published report for a thin AlScN film by more than an order of magnitude[11,12]. To recognize the trend easily, the plots were subdivided and prepared in Supplementary Fig. S7.

Finally, this trend graph also reveals that the relationship between frequency and endurance is influenced by the magnitude of the conventional $2P_r$, which is maintained close to preset $2P_r$. Generally, higher frequencies result in increased endurance performance[29,30]. However, our results show different tendencies depending on the conventional $2P_r$ due to the self-adjusted voltage. For larger conventional $2P_r$ values, endurance tends to decrease with increasing frequency, indicating an inverse correlation. In contrast, at lower conventional $2P_r$ values, such as 10 μC/cm², endurance improves as frequency increases, as shown in Supplementary Fig. S8. Additionally, the general trend of endurance dependence on the frequency can be observed from our test results, as discussed in Fig. 2.

There exists a trade-off between the higher operating voltage and shorter pulse duration as frequency increases. At large preset $2P_r$, voltage-induced stress dominates and reduces endurance, whereas at small preset $2P_r$, the reduced stress exposure time becomes more significant, making endurance effectively frequency independent. The results observed above can therefore be explained by this competing relationship. As frequency increases, $V_C$ also increases, as shown in Supplementary Fig. S3 and Fig. S9. Consequently, regardless of the preset $2P_r$, achieving a conventional $2P_r$ close to the preset $2P_r$ value requires a higher applied voltage at higher frequencies. Such an increase in applied voltage results in two negative effects. Increased applied voltage introduces additional electrical stress, accelerating dielectric breakdown due to a large electric field[31]. Furthermore, it raises peak current density, leading to increased electromigration and Joule heating, accelerating material degradation again[32,33]. These negative effects of frequency reduce endurance, particularly at high

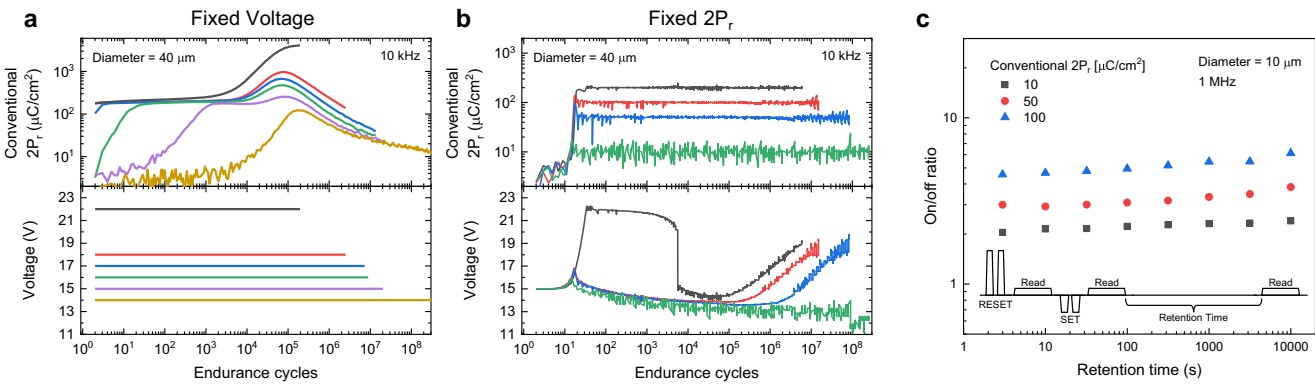

**Fig. 2 | The results of endurance and retention test. a** Endurance under constant applied voltage pulses and **b** adjusted applied voltage pulses to maintain a constant conventional 2P$_r$. **c** Retention performance of partial polarization, with the inset cartoon showing the voltage configuration used in the test.

preset 2P$_r$, which requires a higher applied voltage and leads to increased current density.

However, high frequency increases not only the applied voltage but also the breakdown voltage ($V_{BD}$). At high frequency, pulse width time decreases. This shortens the duration of electrical stress exposure. As a result, the $V_{BD}$ increases. Moreover, a reduction in the preset 2P$_r$ leads to a relatively lower applied voltage at a given frequency. This results in a higher ratio of $V_{BD}$ to applied voltage, reducing electrical stress and limiting dielectric breakdown. Additionally, this lower applied voltage decreases the peak current density, minimizing electromigration and Joule heating significantly. According to previous reports[32], decreases in current density induce dramatic suppression of these negative effects. This is because current based stress generally has a positive correlation with the square of the current. Therefore, these mechanisms enhance endurance, as demonstrated in Supplementary Fig. S8b. The improvement is particularly evident in smaller devices, where reduced defect density further decreases sensitivity to electrical stress. A later section of this paper provides further insights into this effect. These results highlight the importance of optimizing frequency to balance its competing effects on endurance, particularly concerning partial polarization switching.

Figure 2 presents a detailed analysis of endurance characteristics under varying voltage conditions. Figure 2a provides the results obtained from endurance tests conducted at fixed voltages ($V_{fixed}$). The bottom panel of the figure represents the applied voltage, while the top panel shows the evolution of conventional 2P$_r$ throughout the testing cycles. Consistent with prior studies[11,34–38], the wake-up effect was observed in our AlScN capacitors, characterized by an increase in conventional 2P$_r$ after repeated applied voltage cycling. This phenomenon is followed by progressive fatigue leading to device failure. Most conventional endurance tests typically apply voltages exceeding $V_C$. Therefore, most results show a reduction in conventional 2P$_r$, which continues to decline over the last one to three decades of the endurance test. This degradation raises concerns about the reliability of ferroelectricity and suggests that endurance values may be overestimated. However, our study employs a wide range of $V_{fixed}$ for partial and full switching. This approach explores the potential of partial polarization in enhancing device endurance and mitigating degradation. The black curve in Fig. 2a corresponds to a voltage which is much higher than $V_C$, exhibiting behavior consistent with high-performance ferroelectric devices reported in previous literature[29,34]. This includes a short wake-up phenomenon and a gradual degradation, similar to previously observed trends[11,34–38]. These consistent results reinforce the reliability of our endurance measurements and the high quality of our AlScN. Moreover, further results of endurance cycles under various frequencies are shown in Supplementary Fig. S10. Additionally, endurance at 22 V across different frequencies (shown in

Fig. 2a and Supplementary Fig. S10) shows explicitly the same positive correlation observed in previous reports[29,30].

To investigate the influence of partial switching on endurance, progressively lower voltages were applied, as indicated by the colored curves in Fig. 2a. The simple flowchart used for testing the $V_{fixed}$ endurance test is described in Supplementary Fig. S6. These tests were performed with a constant applied voltage until breakdown. A clear trend emerges where lower voltage correlates with enhanced endurance, highlighting the potential reliability benefits of partial polarization. For example, the purple curve exhibits minimal initial conventional 2P$_r$, indicating suppressed polarization switching at the early stages due to the small amplitude of the applied voltage pulse. This endurance behavior can be divided into four distinct phases. This progression of conventional 2P$_r$ evolution is important and requires further discussion. For a comprehensive analysis, refer to Supplementary Fig. S11 and Supplementary Note 5. However, a brief description of each phase follows. The first phase of the $V_{fixed}$ test begins with the onset of polarization switching, characterized by a gradual increase in conventional 2P$_r$. The first phase of the $V_{fixed}$ test is the result of the partial wake-up phenomenon. This is followed by the second phase of the $V_{fixed}$ test, where conventional 2P$_r$ remains constant at a stable value. Subsequently, the third phase of the $V_{fixed}$ test is marked by a pronounced increase in conventional 2P$_r$, which is progressively overestimated due to increasing leakage current. Conventional 2P$_r$ is overestimated because trap-induced relaxation tails make N include more leakage than D. This asymmetry, amplified by defect-mediated relaxation and polarization-dependent leakage, artificially raises the measured conventional 2P$_r$. This is described in Supplementary Note 5. Finally, the fourth phase of the $V_{fixed}$ test is characterized by a reduction in conventional 2P$_r$ due to degradation of the ferroelectricity. The fourth phase of the $V_{fixed}$ test is consistent with the fatigue phenomenon reported in previous studies[11,34–38]. Even under sub-coercive voltages, the wake-up effect remains evident[39]. The presence of a partial wake-up phase (first phase of the $V_{fixed}$ test) delicately activates the device without causing unnecessary high electrical stress. This controlled activation contributes to prolonged endurance performance. Degradation of AlScN begins when increased leakage is initiated. However, it progresses at a slower rate compared to devices subjected to higher voltages. These observations suggest that fine-tuning the applied voltage can significantly enhance the endurance and operational reliability of ferroelectric devices.

In Fig. 2b, we maintained consistent conventional 2P$_r$ values during endurance tests by carefully modulating the applied voltage. This approach provides several advantages. First, it introduces a novel method for controlling polarization in ferroelectric materials, enabling broader utilization of AlScN. Fine-tuning the applied voltage enhances adaptability for industrial applications. This approach makes AlScN

more suitable for applications demanding different $2P_r$ values from a single material. Second, this method ensures high reliability by stabilizing $2P_r$ values without fluctuations throughout operation. This stability helps engineers design circuits with a clear understanding of the behavior of AlScN. This approach prevents performance degradation and extends device lifespan. Finally, we achieved significant improvements in endurance cycles. Our approach maintains stable conventional $2P_r$ throughout the test duration, demonstrating consistent performance over extended cycles.

The applied voltage was carefully controlled using an algorithm shown in Supplementary Fig. S6, which illustrates the programming flowchart implemented with the Keithley Kult software. To keep the conventional $2P_r$ response stable, the evolution of voltage is divided into four phases. More details can be found in Supplementary Fig. S11 and Supplementary Note 5. During the initial phase of the Fixed $2P_r$ ($2P_{r\_Fixed}$) test, the starting voltage cannot be predetermined. To raise the conventional $2P_r$ response to approach the preset $2P_r$, the applied voltage was carefully increased, while the algorithm ensured a rapid rise within tens of cycles to minimize stress. This was followed by the second and third phases of the $2P_{r\_Fixed}$ test, which exhibit distinct characteristics depending on the preset $2P_r$ value. For instance, in the case of the black curve in Fig. 2b, the second phase of the $2P_{r\_Fixed}$ test shows a rough decrease in voltage following an initial increase. Subsequently, in the third phase, a sharp voltage drop is observed, followed by a continued gradual decline. These trends align with the behaviors observed in the second (stable) and third (leakage) phases of the $V_{fixed}$ test (detailed in Supplementary Fig. S11 and Supplementary Note 5). In contrast, for lower preset $2P_r$ values, the sharp voltage drop is not apparent. Instead, the second and third phases of the $2P_{r\_Fixed}$ test progress simultaneously at a slower rate due to the partial wake-up process of the ferroelectric system. During this process, both partial wake-up and stabilization emerge together. The gradual rise in adjusted applied voltage in AlScN contributes to enhanced endurance by mitigating stress-induced degradation and ensuring long-term operational reliability. Notably, as preset $2P_r$ decreases, the distinction between the partial wake-up and stabilization phases becomes increasingly indistinct, as illustrated in Supplementary Fig. S11c. As the device approaches the end of its operational lifespan, the applied voltage is incrementally increased to sustain the conventional $2P_r$ response, a period corresponding to the fourth phase of $2P_{r\_Fixed}$ (fatigue phase). The feedback loop is designed to keep the conventional $2P_r$ response as close as possible to the preset $2P_r$, as shown in Supplementary Fig. S6a. Adjustment sensitivity parameters can be controlled by setting the error threshold ($Error_{th}$) parameter and tuning various parameters into the optimization process. These parameters are configured before the test begins. Moreover, the pulse configuration parameters for the PUND and fatigue pulses are configured together. (Detailed description in Supplementary Fig. S6).

In Fig. 2c, the retention was tested using the pulse sequence illustrated in the inset of Fig. 2c. Retention was evaluated by first presetting partial polarization using repeated RESET/SET pulse pairs until the desired $2P_r$ level was reached, followed by a wait period and a low-voltage read to minimize disturbance. The retention state was quantified by comparing the read currents before and after the wait time ($I_{on}/I_{off}$), where no significant change indicates stable partial polarization. This result agrees with earlier studies that also reported strong retention in partially switched AlScN[23]. In Supplementary Fig. S12, additional retention measurements further confirmed that partial polarization remains stable across different device sizes and frequencies.

Figure 3 exhibits the dependence of ferroelectric behavior in AlScN capacitors on the top electrode diameter. In Fig. 3a, the relationship between electrode diameter and two critical parameters is shown. The parameters under investigation are the coercive ($E_C$) and

breakdown electric fields ($E_{BD}$). These critical values are divided by the 45 nm thickness of the AlScN layer, corresponding to the $V_C$ and $V_{BD}$. $V_C$ values were extracted from PUND measurements (refer to Supplementary Fig. S13). The $E_C$ values exhibit negligible variation across different electrode diameters, suggesting stable polarization switching characteristics. On the other hand, $E_{BD}$ exhibits a significant increase as the electrode diameter decreases[9]. Smaller electrode dimensions lead to a higher $E_{BD}/E_C$ ratio, which is positively linked to improved endurance in ferroelectric capacitors. The increase in $E_{BD}$ with smaller electrode sizes is primarily due to a lower chance of defects forming conductive paths between the top and bottom electrodes. While the intrinsic defect density within the capacitor remains constant, the total number of defects capable of initiating breakdown diminishes with a reduction in electrode area[40]. This decrease in potential failure sites contributes to improved device reliability and extended operational endurance[41]. In Supplementary Fig. S14, breakdown statistics were further analyzed using a Weibull model, which confirms weakest-link scaling with electrode size and reveals consistently high β values, indicating uniform intrinsic reliability. This is a consistent result with a previous report[42]. Our observations suggest that reducing the electrode diameter allows for better defect control[40], which in turn directly influences the breakdown performance[41]. This confirms that fewer defects improve breakdown behavior, reinforcing that smaller electrodes enhance endurance. Moreover, we repeated the measurements at different voltage frequencies (refer to Supplementary Figs. S3 and S9). As reported in previous research[6,9,27,28], the $E_C$ increased with frequency. Our measurements also show that $E_{BD}$ increases with frequency, consistent with a previous report[14].

Furthermore, Supplementary Fig. S15 presents a detailed analysis of the relationship between conventional $2P_r$ and $E_C$ or adjusted applied voltage, with variations in electrode diameter and frequency. This comparison highlights key trends in switching characteristics, offering insights into the effects of scaling and frequency dependence in AlScN capacitors.

In Fig. 3b, the data from Fig. 1f is reorganized to focus on the relationship between electrode diameter and maximum endurance cycles. As expected, the maximum endurance cycles increased as the diameter decreased, which corresponds with the trends observed in Fig. 1f. This suggests that there is room for additional improvements in endurance. To further investigate this behavior, additional reorganized plots were prepared in Supplementary Fig. S16. The endurance enhancement with reduced diameter follows the trend observed across all frequency conditions.

Detailed endurance outcomes for 10 μm diameter capacitors, representing both the lowest (preset $2P_r = 10\ \mu C/cm^2$) and highest (preset $2P_r = 200\ \mu C/cm^2$) maintained conventional $2P_r$ values, are presented in Fig. 3c and d, respectively. These figures illustrate two representative examples of 10 μm diameter capacitors that exhibit different conventional $2P_r$ values. Notably, the capacitor maintaining a conventional $2P_r$ of $10\ \mu C/cm^2$ at 10 kHz demonstrates endurance exceeding 1 billion cycles, underscoring the exceptional performance of AlScN capacitors. For plots of the PUND current response under different conditions measured at intermediate points during the endurance tests, please refer to Supplementary Figs. S17 and S18.

Figures 3c and d display not only conventional $2P_r$ but also charge density (P & N) on the y-axis. Conventional PUND methods, such as subtracting U from P or D from N, underestimate the $2P_r$ at low voltages due to incomplete domain switching. This is because some dipoles must still be partially switched during the U and D pulses (the second pulse of each polarity). We define P and N, divided by area (P/A and N/A) as the 'upper bound $2P_r$'. This method may slightly overestimate $2P_r$ due to leakage currents or RC delays. However, these effects are minimal at low applied voltages. Therefore, in Fig. 3c, the upper bound $2P_r$ of approximately $34.25\ \mu C/cm^2$ ensures high

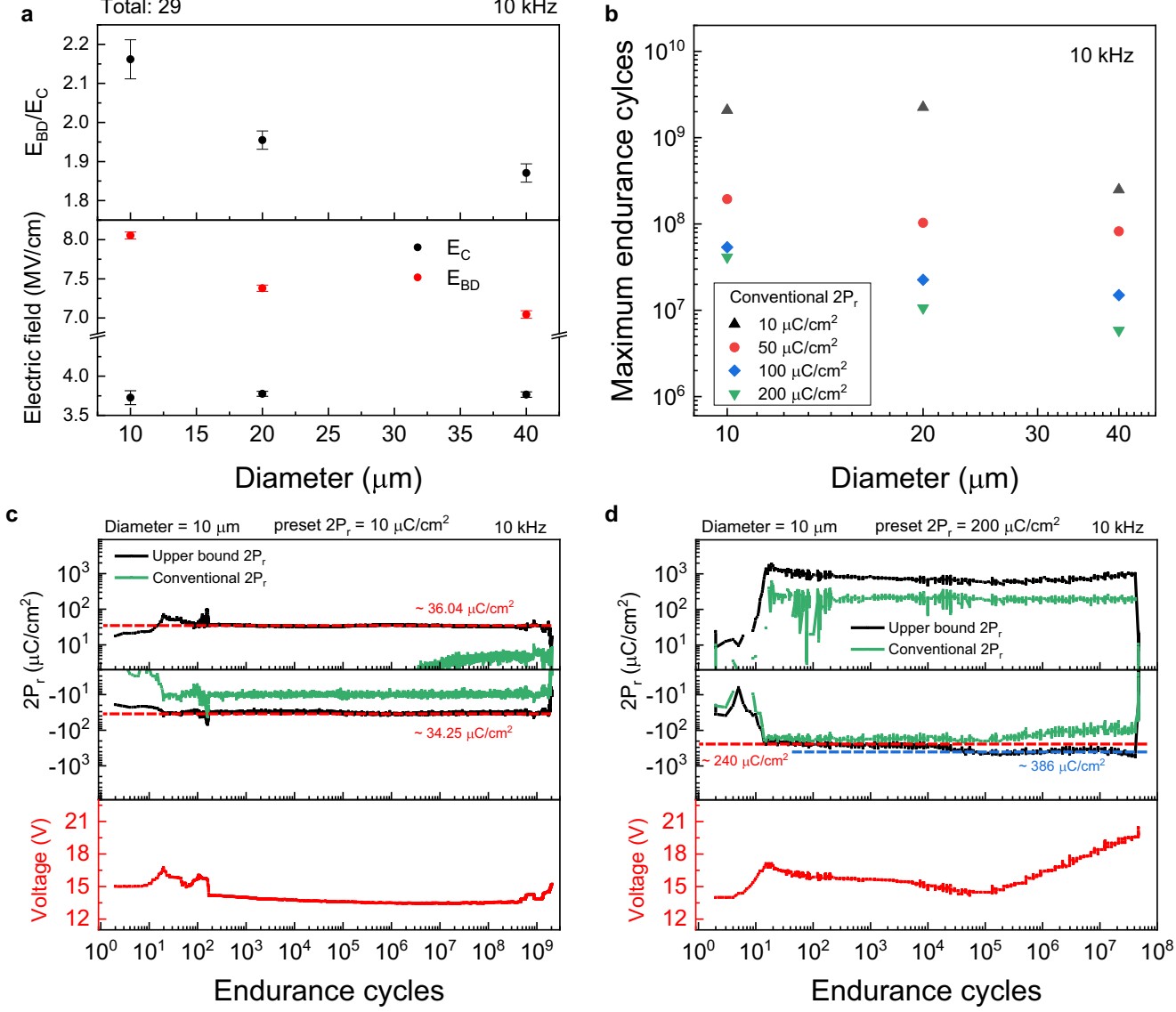

**Fig. 3 | Dependence of Ferroelectric behavior on electrode diameter in AlScN capacitors. a** shows the relationship between electrode diameter and ferroelectric characteristics such as $E_C$, $E_{BD}$ (bottom panel), and their ratio (top panel). Error bars represent the standard error of the mean. **b** displays the trend of endurance cycles as a function of electrode diameter. **c, d** represent endurance test results corresponding to preset $2P_r$ values of 10 μC/cm² and 200 μC/cm², respectively, with their upper bound $2P_r$ values of 34 μC/cm² and 240 μC/cm².

measurement fidelity. On the other hand, the fully switched intrinsic $2P_r$ requires relatively higher voltages, increasing the risk for overestimating the switched polarization. However, before the AlScN fully enters the leakage or degradation phases, this overestimation is much lower. This is because the relatively low voltage applied throughout the entire test is enough to obtain the conventional $2P_r$, reaching the preset $2P_r$. Before the leakage and degradation phase, the upper bound $2P_r$ (-240 μC/cm²) approaches the expected value (200 μC/cm² of conventional $2P_r$), indicating that leakage and other overestimation factors are minimized. This observation confirms the excellent quality of our AlScN. This finding shows that our AlScN capacitor exhibits minimal leakage and few defects due to optimized sputtering and carefully designed capacitors. This reflects excellent stability and outstanding material quality. Further details and analyses are in Supplementary Note 4. Additionally, Supplementary Fig. S19 presents our best endurance test results. This plot highlights the remarkable durability of our device, further validating the robustness of our findings.

In Fig. 3c, we observed that the conventional $2P_r$ under positive bias exhibits a negative value during the early stage of the endurance test. This response is attributed to the slow switching dynamics of partial polarization. Additional details are provided at the end of Supplementary Note 2.

We compare the endurance performance of our AlScN with previously reported ferroelectric materials. Here, we use the conventional $2P_r$ for consistency in comparison with previous reports. As shown in Fig. 4a, our results cover a wide range of $2P_r$ values, demonstrating a unique capability that has not been previously reported. Unlike prior studies, which did not demonstrate controllable $2P_r$, our findings allow comparisons not only within similar $2P_r$ regimes but also with other ferroelectric materials such as PZT and HfO₂-based ferroelectric materials. In addition, the intrinsically high $2P_r$ level of AlScN enables us to explore various $2P_r$ states. A lower intrinsic maximum $2P_r$ level would severely limit the ability to access multiple $2P_r$ states. Our results also confirm that our endurance properties surpass those of other AlN-based ferroelectrics. Although the endurance performance at low $2P_r$

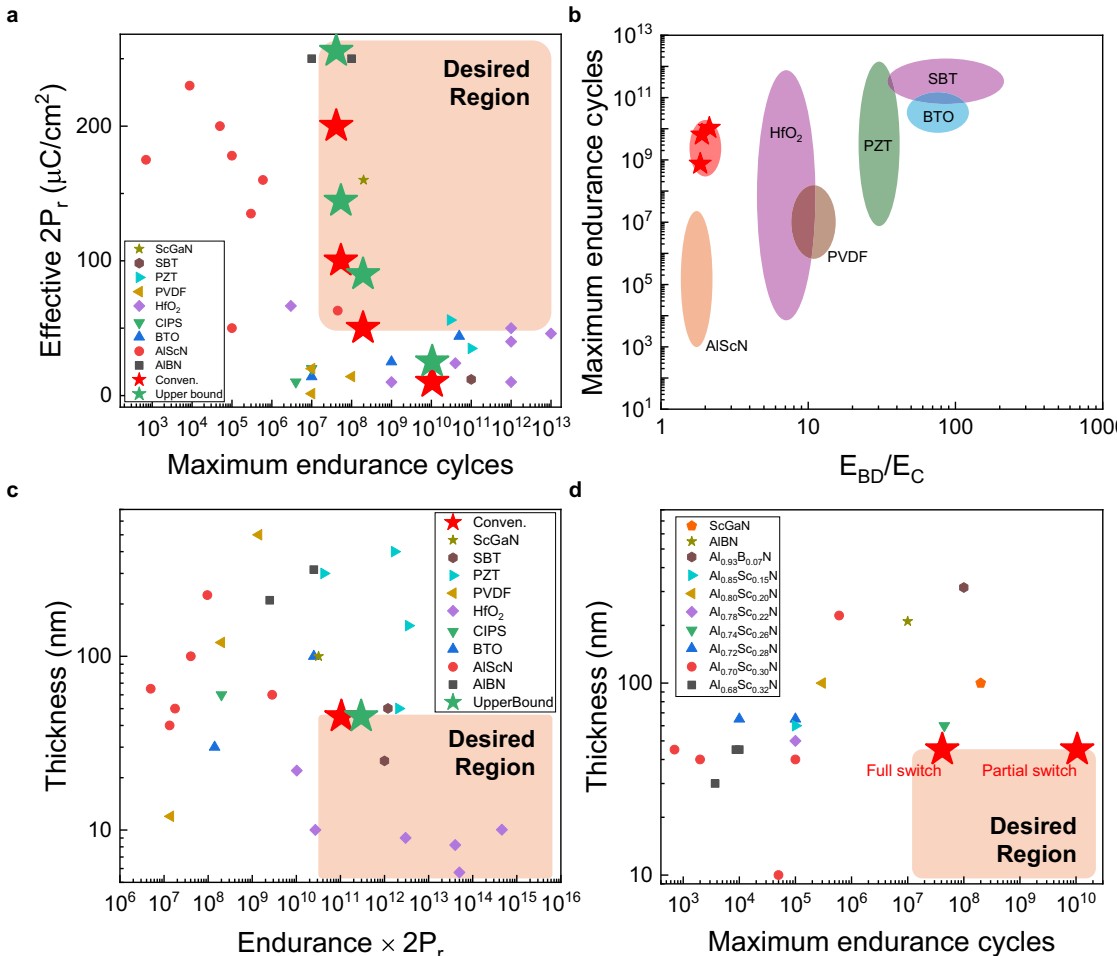

**Fig. 4 | Benchmarking of ferroelectric endurance performance.** A systematic comparison of AlScN endurance characteristics with previously reported ferroelectric materials. The red star markers represent the conventional $2P_r$ and the green star markers represent the value of the upper bound from this study. Among the multiple data points collected under identical x-axis conditions, the most representative dataset was selected. Benchmarking maximum endurance cycles against effective **a** $2P_r$ and **b** $E_{BD}/E_C$ ratio. **c** Benchmark of normalized endurance, obtained by multiplying endurance by $2P_r$ for a reasonable comparison against thickness across different ferroelectric materials. **d** Thickness-dependent endurance benchmarking among Nitride-based ferroelectrics. Data from: AlScN ($Al_{1-x}Sc_xN$[4,11,14,17,35,36,44–50]), AlBN ($Al_{1-x}B_xN$[39,51]), ScGaN ($Sc_xGa_{1-x}N$[52]), HfO2[43,53–59], PZT ($Pb(Zr_xTi_{1-x})O_3$[60–63]), PVDF (poly(vinylidene fluoride)[64–66]), BTO ($BaTiO_3$[67–70]), CIPS ($CuInP_2S_6$[71,72]), SBT ($SrBi_2Ta_2O_9$[73,74]).

remains below the best-reported values, it remains competitive. Moreover, as mentioned above, further performance enhancements are possible. AlScN is emerging as a stronger candidate than traditional ferroelectric materials. The y-axis of effective $2P_r$ represents the most dominant $2P_r$ value observed throughout the endurance test, consistent with previous reports[43].

As illustrated in Fig. 4b, our data follows the trend showing that increased $E_{BD}/E_C$ ratio corresponds to increased endurance. In this comparison plot we can find a significant enhancement in both parameters. The $E_{BD}/E_C$ ratio is higher than in previous reports about AlScN, indicating superior material quality. Moreover, endurance improvement is nearly 2–3 decades greater than previously reported values of AlScN. In Fig. 4c, the x-axis is normalized to provide a clearer evaluation metric. Normalized endurance is an important factor in evaluating ferroelectric materials, as endurance and polarization are often inversely correlated. High polarization accelerates fatigue due to excessive domain wall motion and defect activation, whereas low polarization extends endurance while limiting overall ferroelectric performance. To balance this trade-off, multiplying endurance by $2P_r$ provides a more reasonable comparison metric. Despite these challenges, our results demonstrate significantly superior performance compared to other AlScN studies. In the next step, scalability emerges

as the central consideration. Although our devices demonstrate outstanding performance, accessing the targeted operational window will necessitate further reduction of thickness toward sub-10 nm. Ferroelectric switching has been confirmed at this scale[9], yet systematic endurance evaluations remain scarce. At such dimensions, endurance is expected to be governed by the interplay of coercive and breakdown fields together with leakage and depolarization. Establishing reliable benchmarks in this regime will be essential to validate the long-term applicability of AlScN for advanced memory technologies. For a more detailed comparison, in Fig. 4d, each data point for AlN based ferroelectric materials is scattered in a single plot. Our study presents substantial advantages in terms of both thickness and endurance. Even for fully switched $2P_r$ endurance results, our results establish new benchmarks. Additionally, when partial polarization is considered, significant improvements of approximately three orders of magnitude higher than previous records are observed. These findings position AlScN as a transformative material in ferroelectric endurance research.

We further analyzed the power consumption of our devices in comparison with HfO2-based ferroelectrics. As shown in Supplementary Table S1, the per-cycle energy of AlScN is generally higher due to its large polarization and coercive field. However, partial polarization switching substantially lowers the energy cost, indicating the

possibility of achieving efficiency comparable to Hf-based systems under optimized conditions.

Our work establishes a new paradigm for achieving ultra-high endurance in wurtzite nitride ferroelectric devices through controlled partial polarization switching. By demonstrating write cycling endurance exceeding $10^{10}$ cycles in AlScN—a thousand-fold improvement over previous benchmarks—we overcome a fundamental limitation in wurtzite ferroelectric reliability while maintaining switched $2P_r > 30 \, \mu C/cm^2$. The combination of partial polarization control and device scaling not only extends endurance but also enables operation at reduced voltages, addressing both reliability and energy efficiency challenges. These findings reveal that wurtzite ferroelectrics can surpass the endurance-polarization trade-off traditionally seen in oxide ferroelectrics, opening new opportunities for practical nonvolatile memories. Beyond memory applications, our approach of controlled partial switching provides a general strategy for enhancing reliability in other emerging ferroelectric materials and devices. This work transforms our understanding of polarization dynamics in wurtzite structures while establishing engineering principles for next-generation ferroelectric technologies.

## Methods

### Substrate preparation and $Al_{0.64}Sc_{0.36}N$ deposition

The fabrication process began with the deposition of an $\mathbf{Al_{0.64}Sc_{0.36}N}$ thin film on a 6-inch sapphire wafer with a C-plane (0001) orientation off M-plane (1–100) by $0.2 \pm 0.1$ degrees. A 50 nm Al layer was first deposited at 150 °C. A 45 nm AlScN layer was then co-sputtered at 350 °C with a nitrogen ($N_2$) flow of 30 sccm without argon (Ar) gas, leading to a process pressure of $1.3 \times 10^{-3}$ mbar, utilizing 900 W and 700 W power for 100 mm diameter Al and Sc targets, respectively. Finally, a 50 nm aluminum layer was deposited at 150 °C as the top capping layer. The chamber base pressure was maintained at $4.0 \times 10^{-8}$ mbar. The entire deposition process was conducted in situ to prevent oxidation and maintain film integrity using an Evatec Clusterline 200 II system using high-purity Al (99.999%) and Sc (99.99%) targets. This controlled deposition ensured a high-quality crystalline structure with minimal defects. The composition of $\mathbf{Al_{0.64}Sc_{0.36}N}$ was carefully optimized to balance $P_r$ and $E_C$, ensuring robust ferroelectric behavior.

### Device fabrication

The first electron beam lithography (EBL) process was performed to define a pattern facilitating via contact formation while preserving the top Al layer using an EBPG5200 + , Raith. A 50 nm thick chromium (Cr) layer was deposited using an E-Beam sputtering system (PVD 75, Kurt J. Lesker) to protect the top Al electrode. A second EBL step was employed to pattern the top electrode structure, followed by inductively coupled plasma (ICP) etching to selectively remove the top Al layer using a Cobra PlasmaPro 100, Oxford Instruments. This Al layer, originally serving as a capping layer, remained functional as the top electrode, minimizing interfacial defects between AlScN and the top Al electrode. The AlScN film and the bottom electrode were subsequently etched using the same EBL and ICP etching process. A 200 nm thick $SiO_2$ insulating layer was deposited using plasma-enhanced chemical vapor deposition (PECVD) using a PlasmaLab 100, Oxford. Following this, via contacts and contact pads were patterned by an EBL process. The etching process was conducted by a Reactive-ion etching (RIE) process using an 80 Plus, Oxford Instrument. Finally, to fill the via hole and define the contact pad, Pd/Ti were deposited using an E-beam evaporation process.

### Electrical characterization measurement

The electrical characterization was carried out at room temperature using a probe station interfaced with a Keithley 4200A-SCS semiconductor parameter analyzer (Tektronix Inc.).

## Data availability

The data that support the findings of this study are present in the Article and the Supplementary information or available from the corresponding authors upon request.

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

## Acknowledgements

The authors acknowledge support from the Intel SRS program. D.J. also acknowledges partial support from the Office of Naval Research (ONR) Nanoscale Computing and Devices program (N00014-24-1-2131) and the Air Force Office of Scientific Research (AFOSR) GHz-THz program grant number FA9550-23-1-0391. D.J. also acknowledges partial support from NSF Future of Semiconductors (FuSe) program ECCS 2328743. A portion of the sample fabrication, assembly, and characterization were carried out at the Singh Center for Nanotechnology at the University of Pennsylvania, which is supported by the National Science Foundation (NSF) National Nanotechnology Coordinated Infrastructure Program grant NNCI-1542153. The authors acknowledge the use of an X-ray diffraction facility supported by the Laboratory for Research on the Structure of Matter and the NSF through the University of Pennsylvania Materials Research Science and Engineering Center (MRSEC) DMR-2309043.

## Author contributions

D.J. and R.H.O. conceived the idea and designed the overall experiments. H.C. developed the code for the endurance cycle test. H.C., Y.W., Y.H., and Z.H. conducted the current-voltage measurements. R.H.O. supervised the AlScN growth process. H.C., C.L., and Y.Z. deposited the AlScN. H.C. designed and carried out the device fabrication processes. X.T. and J.T. conducted the XRD measurement and analysis. H.C. and V.D.B. conducted the PFM measurement and analysis. D.J., R.H.O., and H.C. analyzed the data, prepared the figures, and wrote the manuscript. All authors contributed to the discussion, analysis of the results, and manuscript writing.

## Competing interests

The authors declare no competing interests.
