## [Transparent Peer Review file · Nature Communications]

Write Cycling Endurance Exceeding 10^{10} in Sub-50 nm Ferroelectric AlScN

Corresponding Author: Professor Deep Jariwala

Version 0:

Reviewer comments:

Reviewer #1

(Remarks to the Author)

The authors present a carefully curated study on AlScN-based thin films where they test the device reliability/endurance by cycling up to 10^{10} cycles. Here are a few suggestions which can help improve the readability and reproducibility of the manuscript:

(a) In Main section, paragraph 1/ Fig. 1 a, the authors mention: "an Al_{0.64}Sc_{0.36}N capacitor specifically designed to minimize stress on the Al_{0.64}Sc_{0.36}N layer during measurements. This reduction in stress was achieved to avoid the pressure of the probe tip with the via contact."

The authors should consider either providing literature-based evidence supporting this claim or including appropriate endurance data for a more standard version (not a via-contact structure) of a ferroelectric-based capacitor heterostructure with patterned top electrode/blanket ferroelectric film/bottom blanket electrode film on the same substrate.

(b) In Fig. 1f, the authors mention that with increasing the endurance cycles, the 2Pr value changes from conventional (~200 $\mu\text{C}/\text{cm}^2$) to a reduced value of ~10 $\mu\text{C}/\text{cm}^2$ due to increased coercive voltage. The authors mention:

"As the frequency increases, an increase in operation voltage was observed. The increase of operation voltage refers to the applied voltage necessary to obtain the same conventional 2Pr. Additionally, a difference in the slope of conventional 2Pr from onset to saturation was detected. This indicates a distinct frequency-dependent modulation of domain switching. The observation of increased coercive voltage (VC) is consistent with reports in previous literature^{6,9,13,24,25}. This suggests that the switching kinetics are influenced by the interplay between external field dynamics and the intrinsic material response. This interplay highlights the importance of further exploring frequency-dependent polarization behavior."

From the text mentioned by the authors, the change in modulation of domain switching (and hence 2Pr/Vc) is apparent as a function of changing frequency. It is, however, not clear to me why the 2Pr value changes with the increase in endurance cycles which is expected to have been conducted at the same frequency in PUND sequence? The authors should elaborate on the same.

(c) Was the leakage current monitored as a function of endurance cycles (akin to that in Fig. 1f)? If so, was it consistent in value? A higher magnitude of leakage current at higher endurance cycles could lead to a lower 2Pr value therein. The authors should discuss such possibilities with data or literature-based evidence.

Reviewer #2

(Remarks to the Author)

AlScN has emerged as a promising materials platform for non-volatile memories due to its high polarization and compatible process of back-end-of-line. However, it still shows limited endurance, typically failing before 10^7 switching cycles. In this article, the authors demonstrate unprecedented control of polarization switching in AlScN, achieving write cycling endurance exceeding 1010 cycles in 45 nm-thick Al_{0.64}Sc_{0.36}N capacitors.

While the high endurance of AlScN is obtained by controlling the partial switching, we do not think the study offers sufficient novelty to publication in Nature Communications due to lacking the comprehensive fatigue mechanism. Below are detailed comments and suggestions.

1. The basic materials characteristic should be supplied, such as, XRD, PFM
2. The electrode has important on the endurance for ferroelectric. How about the endurance of AlScN when using other electrode metal, such as, Sc, Cr?
3. The author claims that the endurance is up to 1010 cycle when partial switching is $30 \mu\text{C}/\text{cm}^2$. How about the retention time of partial switching at $30 \mu\text{C}/\text{cm}^2$.
4. In figure.1f, the conventional 2Pr is $50 \mu\text{C}/\text{cm}^2$, $100 \mu\text{C}/\text{cm}^2$ and $200 \mu\text{C}/\text{cm}^2$, the endurance increase by increase frequency. When conventional 2Pr is $10 \mu\text{C}/\text{cm}^2$, the endurance has no relationship with the frequency. Why?
5. Is this method general for different thickness AlScN?
6. How many devices was conducted in this paper?
7. In this article, the fatigue mechanism should be discussed detailly?

Reviewer #3

(Remarks to the Author)

This study demonstrates an impressive breakthrough in enhancing the write cycling endurance of wurtzite ferroelectric AlScN to over 10^{10} cycles via controlled partial polarization switching and device scaling, representing a thousand-fold improvement over previous benchmarks. The research addresses a critical challenge in ferroelectric memory applications, with well-designed experiments and comprehensive data supporting the findings. While the work is innovative and impactful, several aspects of the mechanism interpretation, comparative analysis, and experimental details could be further refined.

1. The study attributes enhanced endurance to "incomplete domain switching" during partial polarization, but lacks microscopic evidence (e.g., TEM/PFM images) of domain structure evolution. A theoretical model explaining domain wall pinning or defect-mediated switching in AlScN would strengthen the mechanism interpretation.
2. The benchmarking in Fig. 4 omits recent advancements in 2D ferroelectrics (e.g., CuInP₂S₆) and other nitrides (e.g., ScGaN). Including these would more robustly validate AlScN's performance edge.
3. Quantitative comparison of energy consumption per cycle between AlScN and HfO₂-based ferroelectrics is missing, which is critical for memory applications. Discussing this trade-off would enhance practical relevance.
4. The AlScN deposition parameters (e.g., sputtering pressure, target purity) in the Methods section are insufficient for reproducibility. Specify details such as chamber pressure, N₂/Ar flow ratio, and target material specifications.
5. While the study mentions defect density reduction with electrode scaling, a quantitative statistical model (e.g., Weibull distribution) for breakdown probability across different sizes would strengthen the theoretical framework.
6. The effect of partial polarization on read/write speed and retention property, which are important practical metrics, were not discussed. Addressing these via additional tests or theoretical predictions would be valuable.

Reviewer #4

(Remarks to the Author)

In this paper, the authors measured the endurance of AlScN capacitors and claimed to achieve record-high performance. While the measurement and analysis are comprehensive, the paper lacks novelty. The data quality is poor, and the analysis is not sufficiently rigorous. The reported improvement in endurance over previous literature is marginal and does not justify publication in Nature Communications. The following are my questions and comments.

1. In Fig. 1d, the leakage current under positive bias is significantly larger than under negative bias. Please explain the underlying physical mechanism responsible for this asymmetry.
2. The authors use P/area (N/area) values instead of $P-U/\text{area}$ ($N-D/\text{area}$) to estimate the partial polarization. This method can lead to an overestimation of remanent polarization, as the charge from leakage current is also included. Please correct the corresponding remanent polarization values to allow a fair comparison with the literature.
3. In Fig. 2a, the third phase of the fixed-voltage test shows an increase in the conventional 2Pr, which the authors attribute to leakage current. However, in principle, the PUND method should subtract the leakage contribution using two consecutive pulses. Please clarify how leakage current could lead to an apparent increase in the extracted 2Pr.
4. In the endurance test, as the device approaches its end-of-life, the leakage current increases significantly. In this regime, the leakage current is no longer negligible during polarization measurements, resulting in substantial errors in the "intrinsic 2Pr" extraction method. Consequently, the endurance number obtained using this method is not reliable.

5. For Fig. 4, please include plots of both the “upper bound 2Pr” and the “lower bound 2Pr” (conventional 2Pr) to enable a fair comparison with other works.

Version 1:

Reviewer comments:

Reviewer #1

(Remarks to the Author)

The manuscript can be accepted in the current revised state.

Reviewer #2

(Remarks to the Author)

In the revised manuscript, the authors addressed all of my questions with adequate experimental data. I recommend publishing the manuscript in Nature Communications in its current form.

Reviewer #3

(Remarks to the Author)

The author has replied to all my questions.

Reviewer #4

(Remarks to the Author)

It remains unclear what the novelty of this paper is. The device structure is very simple, and the measurement methodology is conventional. The data quality appears noisy, and the reported endurance value is only marginally better than previously published results. It is therefore difficult to justify why this work merits publication in Nature Communications.

The authors have addressed my technical questions; however, these clarifications and revisions do not alleviate the fundamental concerns regarding the novelty and significance of the work.

Reviewer #1 (Remarks to the Author):

The authors present a carefully curated study on AlScN-based thin films where they test the device reliability/endurance by cycling up to 10^{10} cycles. Here are a few suggestions which can help improve the readability and reproducibility of the manuscript:

Response:

We thank the reviewer for the positive evaluation of our work and for recognizing the careful design of the study. The reviewer's feedback highlights the robustness of our endurance results while pointing out the importance of presenting them with greater clarity, readability, and reproducibility. We fully acknowledge these constructive points and have revised the manuscript to improve the clarity of context and the organization of results. Furthermore, we provide detailed responses to each of the reviewer's specific questions to ensure that the revisions directly address the concern raised.

Reviewer's comments:

(a) In Main section, paragraph 1/ Fig. 1a, the authors mention: "an Al_{0.64}Sc_{0.36}N capacitor specifically designed to minimize stress on the Al_{0.64}Sc_{0.36}N layer during measurements. This reduction in stress was achieved to avoid the pressure of the probe tip with the via contact."

The authors should consider either providing literature-based evidence supporting this claim or including appropriate endurance data for a more standard version (not a via-contact structure) of a ferroelectric-based capacitor heterostructure with patterned top electrode/blanket ferroelectric film/bottom blanket electrode film on the same substrate.

Response:

We appreciate the reviewer's feedback on our device configuration. Because AlScN is ferroelectric⁽¹⁾ and inherently piezoelectric material⁽²⁾, externally applied pressure by probe tips can induce an internal potential as an intrinsic response. In addition, when the probe tip applies force, the local contact region may deform, and the mechanical boundary conditions of the capacitor can be modified^(3, 4). These combined effects can lead to non-uniform electric field distributions and localized current crowding. Such conditions may further enhance self-induced Joule heating or electron–lattice collisions, which in turn increase the unintended phenomena such as dielectric breakdown, electromigration, or other

migration phenomena⁽⁵⁾. Consequently, various unexpected degradation effects can trigger abrupt breakdown events. Here, we provide detailed experimental evidence supporting this breakdown behavior.

Figure R1 | AFM topography before and after breakdown of a non-via contact capacitor. (a) Schematic illustration of the $\text{Al}_{0.64}\text{Sc}_{0.36}\text{N}$ capacitor with a non-via contact structure and direct tip-contact measurement configuration. (b) AFM line-scan profile with inset showing the full 3D topography prior to endurance testing. (c, d) AFM line-scan profiles with corresponding 3D topography insets recorded after accidental breakdown, with (d) highlighting the tip-contact region. Dashed lines in each inset indicate the AFM scan-line trajectories.

Figure R1 (a) illustrates a capacitor measured without via contacts (direct probe-tip contact to the top electrode), and **Figure R1 (b)** presents an AFM line scan with a 3D inset obtained prior to any electrical stress (DC-IV, AC-IV, or endurance test). The dashed line indicates the scan trajectory. Before testing, the surface topography is uniform and smooth as shown in **Figure R1 (b)**. By contrast, after an

unintentional breakdown, the contact site exhibits a crater deformation penetrating into the active AlScN layer, as shown in **Figure R1 (c) and R1 (d)**. The crater depth reaches up to approximately 91 nm at its deepest point, indicating that the pit extends through the ferroelectric film and reaches the underlying platinum (Pt) bottom electrode. Such penetration explains sudden device failure, as the pit creates a conductive path that produces a direct short. Consequently, the current rapidly rises to the compliance limit, giving the appearance of a maximum current spike. This outcome underscores the risk associated with non-via measurements.

To further clarify the endurance behavior, additional experiments were performed on capacitors with 40 μm diameter top electrodes under a target preset $2P_r$ of $\sim 200 \mu\text{C}/\text{cm}^2$ at 1 MHz. The endurance characteristics of via- and non-via configurations are summarized in **Figure R2 (a–d)**.

Figure R2 | Electrical comparison between via and non-via contact capacitors. (a) DC-IV of a well contacted device shows stable behavior. This includes not only via contact but also non-via contact under gentle touch. **(b)** DC-IV of a non-via contact under non-gentle touch shows abrupt

failure, while it is rarely observed with via contact. **(c)** Endurance of a via contact capacitor under adjusted voltage pulses that keep conventional $2P_r$ close to $200 \mu\text{C}/\text{cm}^2$. **(d)** Endurance of a non via contact capacitor under the same condition of **(c)**.

For non-via contact tests, maintaining “gentle touch” is essential. The probe tip force on the sample must be kept as low as possible, such that it only touches the electrode lightly without stressing the ferroelectric layer. Under such conditions the AlScN film remains stable and reproducible measurements can be obtained. Indeed, initial DC-IV small voltage range sweeps were used to confirm gentle touch prior to endurance cycling tests for non-via contact capacitor. Despite this precaution, certain non-via capacitors still failed prematurely around 10^4 cycles, as shown in **Figure R2 (d)**. This behavior is attributed to the inherent fragility of the non-via geometry. Even slight external perturbations can affect the gentle touch configuration and accelerate breakdown.

Moreover, as shown in **Figure R2 (c)** and **(d)**, via contact capacitors and non-via contact capacitors under gentle touch display nearly identical switching in the early stage. This confirms that geometry does not significantly alter the intrinsic electrical properties, provided that gentle contact is maintained in the non-via case. Notably, a via capacitor could still endure up to 10^6 cycles with reliable switching across the full voltage range, as shown in **Figure R2 (c)**. This difference highlights that endurance stability depends not only on the intrinsic characteristics of AlScN but also on the robustness of the contact with probe tip scheme.

These results demonstrate that while the polarization switching response is fundamentally similar, the reproducibility and lifetime are highly sensitive to probing configuration and mechanical loading. The robustness afforded by via contacts, together with careful control of contact force, is therefore essential for achieving reliable long term endurance performance.

To further investigate the post-breakdown state of the non-via contact capacitor, we prepared a device in which the top electrode was selectively removed. The corresponding sample was fabricated with position indicators, as shown in **Figure R3**, to facilitate identification of the location where electrical testing had been performed. For efficient removal of the Al top electrode, we employed a dilute HF solution (5 ml of 2% HF dissolved in 5 ml of deionized water). A device that had experienced abrupt failure during electrical cycling was chosen for this analysis, and AFM imaging was conducted following top electrode removal. **Figure R3 (c)** presents the AFM topography of the breakdown site. The square region highlighted by the white dashed box corresponds to the original capacitor footprint.

The black dashed line is drawn across the region contacted by the probe tip to indicate the line-scan profile trajectory. Line-scan spectroscopy along this black dashed line, as shown in **Figure R3 (d)**, revealed a crater penetrating to a depth of approximately 45 nm. Considering that the Al top electrode was fully etched away, this crater extends nearly to the bottom interface of the ~45 nm-thick AlScN layer, thereby indicating a localized dielectric failure.

Figure R3 | AFM topography after accidental breakdown of a non-via contact capacitor following top electrode removal. (a, b) Schematic illustration of the non-via contact capacitor before and after electrode etching. The red region in (b) denotes the polarization-switched area under applied bias. **(c, d)** AFM measurements after breakdown, where the dashed black line in (c) marks the scan trajectory at the tip-contact location, corresponding to the line profile shown in (d).

These observations indicate that excessive probe loading in non via contact configurations can create sharp local stress and trigger early breakdown. In contrast, via contact structures relieve local stress and improve reliability during endurance testing. This analysis underscores the disadvantage of non via contact devices for long cycling studies and supports the use of via contact designs for robust evaluation

of AlScN capacitors.

Besides the benefits discussed above, via structures provide further practical advantages. Via contacts offer a pad area larger than the capacitor itself, which facilitates accurate probe alignment. When the top electrode diameter is below 15 μm , it is not possible to establish a stable connection using a conventional probe tip in our system. In addition, as noted in our main paper, the contact pad is formed above the region without bottom Al electrode or AlScN piezoelectric layer, allowing it to withstand strong pressure without introducing stress into the AlScN. This ensures stable contact during extended endurance measurements lasting up to fourteen days.

In contrast, non-via structures require exceptionally gentle probe touch to avoid stressing the film. Such delicate contact is often disrupted by vibrations or external perturbations, leading to instability. Despite carefully applying gentle contact, premature breakdown frequently occurred at the early stages of endurance testing. These observations emphasize that, beyond the intrinsic ferroelectric properties, structural considerations are also critical, particularly for materials exhibiting unprecedentedly large $2P_r$ values and small thicknesses.

We thank the reviewer once again for this helpful feedback which contributed to improving our publication.

Change to the manuscript:

Additional supplementary figures have been included in **revised Supplementary Information** to show that via contacts mitigate probe stress and improve endurance reliability. **Figures from R1 to R3** were added as **Supplementary Information S1**, showing AFM evidence of a crater formed by sudden breakdown and presenting electrical results that highlight the differences between endurance testing and DC-IV measurements. The relevant discussion is provided in the **revised Manuscript (page 3, lines 19 in Revised Manuscript.)**

Reviewer's comments:

(b) In Fig. 1f, the authors mention that with increasing the endurance cycles, the $2P_r$ value changes from conventional ($\sim 200 \text{ uC/cm}^2$) to a reduced value of $\sim 10 \text{ uC/cm}$ due to increased coercive voltage. The authors mention:

"As the frequency increases, an increase in operation voltage was observed. The increase of operation voltage refers to the applied voltage necessary to obtain the same conventional $2P_r$. Additionally, a difference in the slope of conventional $2P_r$ from onset to saturation was detected. This indicates a distinct frequency-dependent modulation of domain switching. The observation of increased coercive voltage (VC) is consistent with reports in previous literature^{6,9,13,24,25}. This suggests that the switching kinetics are influenced by the interplay between external field dynamics and the intrinsic material response. This interplay highlights the importance of further exploring frequency-dependent polarization behavior."

From the text mentioned by the authors, the change in modulation of domain switching (and hence $2P_r/V_c$) is apparent as a function of changing frequency. It is, however, not clear to me why the $2P_r$ value changes with the increase in endurance cycles which is expected to have been conducted at the same frequency in PUND sequence? The authors should elaborate on the same.

Response:

We thank the reviewer for carefully evaluating our work and providing constructive feedback to improve the manuscript. We also recognize that the wording in our original submission may have caused confusion.

In **Figure 1f**, which is in our main paper, the term ‘conventional $2P_r$ ’ refers to the value extracted using the PUND method. This terminology emphasizes the measurement approach rather than the maximum capacity of the intrinsic polarization of AlScN. It does not represent the fully saturated polarization near $200 \mu\text{C}/\text{cm}^2$, which is often reported in prior studies as the full polarization value. In our main paper, conventional $2P_r$ values below $200 \mu\text{C}/\text{cm}^2$ occur when the applied pulse amplitude is lower than the voltage required to saturate the polarization at the corresponding frequency. Under these conditions, only a fraction of the domains switch. The measured response therefore reflects partial switching of the polarization.

Endurance tests were carried out at a fixed frequency within each set of tests. The variation in conventional $2P_r$ across sets is not due to frequency, but reflects the intentional adjustment of pulse amplitude to control the switched portion of AlScN, which is a behavior widely observed in ferroelectrics including AlScN^(6, 7). At the start of each run, the applied voltage was tuned to achieve a targeted preset $2P_r$ (one of the 10, 50, 100, or $200 \mu\text{C}/\text{cm}^2$) and then actively maintained during the endurance test. This approach ensures the conventional $2P_r$ results in the targeted preset $2P_r$ and enables

a fair comparison of endurance under various conditions for each partial switching test. The purpose of **Figure 1f** in our main paper is to demonstrate the improvement in endurance achievable under partial polarization switching conditions.

The frequency-dependent increase of coercive voltage was characterized separately in **revised Supplementary figures S3 and S9 in revised Supplementary Information**. We have revised the manuscript to resolve the reviewer's concern and clarify this point.

Change to the manuscript:

We have added a further explanation about **Figure 1f** to prevent the type of confusion noted by the reviewer. Specifically, the following clarification was included. "These values represent controlled partial polarization switching sustained throughout the endurance measurement, as further detailed in the following paragraph." (**Page 6, lines from 6 to 8 in Revised Manuscript.**) In addition, the sentence in the manuscript was revised to "Additionally, a difference in the slope of conventional $2P_r$ from onset to saturation was detected, as shown in **Supplementary Figure S3.**" (**Page 5, line 26 in Revised Manuscript.**)

Reviewer's comments:

(c) Was the leakage current monitored as a function of endurance cycles (akin to that in Fig. 1f)? If so, was it consistent in value? A higher magnitude of leakage current at higher endurance cycles could lead to a lower $2P_r$ value therein. The authors should discuss such possibilities with data or literature-based evidence.

Response:

We thank the reviewer for highlighting the role of leakage current in reducing the measured conventional $2P_r$. We monitored leakage as a function of endurance cycles. In our devices, the leakage contribution appears in the U and D pulses of the PUND measurement sequence. As shown in **revised Supplementary Information S5 in the revised Supplementary Information**, the leakage charge during the D pulse, normalized by area, increases steadily at high cycle counts and follows the drop in conventional $2P_r$ when the drive voltage is fixed. It is shown at the final stage of endurance tests in **revised Supplementary Figures S5-1 to S5-3 of the revised Supplementary Information**. The

related phases are Phase III (**revised Supplementary Figures S5-1 and S5-3**) and Phase IV (**revised Supplementary Figure S5-2**). Under constant applied voltage, the growth of leakage indeed lowers the measured conventional $2P_r$. Similar behavior has been reported in other well-established ferroelectric materials such as Hafnium, Perovskite and Wurtzite based materials^(10 - 18). Many studies attribute it to vacancies and traps in ferroelectrics. When defects become excessive, current paths can form through mechanisms such as the Poole–Frenkel effect⁽⁷⁾, causing the switching energy to bypass into leakage rather than concentrating on the domain switching. This bypassed energy is dissipated as Joule heating instead of being used for polarization switching. In addition, these excessive defects hinder domain movement and induce domain pinning^(11 - 14).

However, we observed that the leakage current and conventional $2P_r$ are not always inversely related. At the leakage onset state during the endurance test, an increase in leakage can coincide with a slight rise in conventional $2P_r$. Because Maxwell–Wagner relaxation⁽⁸⁾ drives Curie–von Schweidler relaxation currents⁽⁹⁾ that add a non-switching contribution to the pulse responses. This leakage-related contribution overlaps with the ferroelectric switching peak, artificially increasing the conventional $2P_r$.

Therefore, the relation between leakage and polarization can either overestimate the conventional $2P_r$ or cause permanent damage that leads to a real loss of polarization, depending on the mechanism. Overall these observations point to a complex and nonmonotonic interplay between leakage evolution and polarization switching during cycling.

However, our self-adjustment study did not show any correlation between leakage current and conventional $2P_r$. Although leakage current changes with cycling, the conventional $2P_r$ remains near its target value because we apply a self-adjusting voltage scheme. After every set of cycles, we perform a PUND check and update the pulse amplitude so that the conventional $2P_r$ stays close to the preset value. In the later stages of each run we increase the voltage only as needed to hold the target $2P_r$. This feedback loop preserves the switched polarization until the film approaches breakdown. The algorithm and pulse scheduling are summarized in **Supplementary Figure S6**, and this context directly connects to our approach. In parallel, to isolate true switching from leakage we used the PUND method to cancel the leakage current in the P and N steps by subtracting the U and D responses and to extract the switched charge reliably.

Figure R4 presents four additional endurance datasets together with the corresponding leakage curves shown by the blue plots in the figures. When the preset $2P_r$ is small, as shown in **Figure R4 (a)**,

the leakage remains effectively suppressed even as the endurance approaches its final stage. This behavior is attributed to the reduced electric stress on the AlScN under a low preset $2P_r$. Even as the endurance approaches its final stage and the control loop raises the applied voltage, the blue D/A curve remains nearly constant. This indicates that the leakage current does not increase under these conditions. Here, we observe that adjusting the applied voltage suppresses the formation of unnecessary defects and mitigates both under- or overestimation of $2P_r$, but above all it prevents degradation of the ferroelectric material. Therefore, voltage adjustment is one of the key factors in extending endurance.

Figure R4 | Additional endurance test results with leakage current. (a) 20 μm diameter, preset $2P_r = 10 \mu\text{C}/\text{cm}^2$. **(b)** 10 μm diameter, preset $2P_r = \mu\text{C}/\text{cm}^2$. **(c)** 40 μm diameter, preset $2P_r = 100 \mu\text{C}/\text{cm}^2$. **(d)** 10 μm diameter, preset $2P_r = 200 \mu\text{C}/\text{cm}^2$. Red and blue curves represent the integrated charge density of the N and D pulses divided by area, respectively. Black curve represents the conventional $2P_r$ extracted as the difference between N and D divided by area. Green curve indicates the applied voltage adjusted during endurance test to maintain the conventional $2P_r$ close to the preset $2P_r$ value.

However, when the preset $2P_r$ increases, as shown in **Figure R4 (b – d)** and **Supplementary Information Figure S5-4** the blue curve rises sharply in the later stage of cycling. The higher target $2P_r$ requires a higher applied voltage, which increases device stress and accelerates leakage. In this process, leakage still grows, but the algorithm only raises the voltage by the minimum amount needed to compensate for the degraded $2P_r$. As a result, even at high preset $2P_r$ values, endurance is extended. The applied voltage adjustment restores the electric field across the AlScN layer while remaining below the breakdown limit. It also suppresses defect generation. As a result, the conventional $2P_r$ stays stable until failure begins.

Unfortunately, it is not feasible to present leakage current in the same figure as in **Figure 1f**, because the leakage does not remain constant during endurance testing. As shown in the plots in **Figure R4**, the current evolves continuously with cycling. Therefore, a representation of maximum endurance versus leakage current is not meaningful, since the leakage cannot be represented by a single value for each test condition.

We sincerely thank the reviewer for the constructive comments on the role of leakage current and its impact on $2P_r$. These insights allowed us to clarify our discussion and place our data on firmer theoretical ground.

Change to the manuscript:

We have added a detailed explanation of the relationship between leakage current and conventional $2P_r$. Based on this understanding, we also provide theoretical support showing that voltage adjustment can suppress degradation and thereby extend the endurance cycles in the **revised Supplementary Information (page 50, lines 4 to 16)**.

Reviewer #2 (Remarks to the Author):

AlScN has emerged as a promising materials platform for non-volatile memories due to its high polarization and compatible process of back-end-of-line. However, it still shows limited endurance, typically failing before 10⁷ switching cycles. In this article, the authors demonstrate unprecedented control of polarization switching in AlScN, achieving write cycling endurance exceeding 10¹⁰ cycles in 45 nm-thick Al_{0.64}Sc_{0.36}N capacitors.

While the high endurance of AlScN is obtained by controlling the partial switching, we do not think the study offers sufficient novelty to publication in Nature Communications due to lacking the comprehensive fatigue mechanism. Below are detailed comments and suggestions.

Response:

We sincerely thank the reviewer for the thoughtful evaluation and for highlighting both the promise of AlScN and the importance of understanding fatigue phenomena. In essence, the reviewer's comment raises the question of how the novelty of endurance achievement relates to the broader mechanistic picture of fatigue. We respectfully acknowledge that fatigue mechanisms are a critical topic in ferroelectric research, and we value the reviewer's perspective on this point. Our intention in this work was to provide a careful experimental foundation for the endurance results, which we hope will be of value for ongoing efforts toward the development of reliable ferroelectric devices. In addition, we provide detailed responses to the specific questions raised by the reviewer.

Reviewer's comments:

1.The basic materials characteristic should be supplied, such as, XRD, PFM

Response:

We appreciate the reviewer's comments on the basic material characteristics, such as X-Ray diffraction (XRD) and piezoresponse force microscopy (PFM). For XRD measurements, the top Al electrode was completely removed. The sample was immersed in a dilute HF solution prepared by mixing 5 ml of 2 % HF with 5 ml of deionized water for 70 seconds. This wet etch exposed the AlScN surface while preserving the Al bottom electrode and the sapphire substrate. XRD was performed on a Rigaku SmartLab SE. Copper K alpha (Cu K α) radiation with a wavelength of 1.540 Å. The two-theta

(2θ) range was 20° to 80° with a step size of 0.01° and a scan speed of 3° per minute.

Figure R5 (a) shows the $\theta/2\theta$ XRD scan data. The scan reveals the $\theta/2\theta$ peaks corresponding to $\text{Al}_{0.64}\text{Sc}_{0.36}\text{N}$ (0002), Al (111), and Sapphire. The AlScN peak at 36.22° is consistent with previous reports, confirming the high quality of our 45 nm AlScN ferroelectric film. The exclusive presence of the (0002) peak indicates that most grains have the c-axis oriented AlScN⁽¹⁹⁻²⁴⁾. From the presence of the Al (111) peak, we confirm that the Al bottom electrode offers a favorable template for the growth of c-axis oriented AlScN^(19, 21, 25). The absence of additional AlScN reflections such as (1010) or (1011) further supports single orientation^(22, 26). In addition, using a single effective wavelength of 1.540 \AA the (0002) peak gives d_{0002} of 2.477 \AA and c of 4.954 \AA which is consistent with other reports^(19, 22).

Figure R5 | XRD of $\text{Al}_{0.64}\text{Sc}_{0.36}\text{N}$ on Al/Sapphire substrate. (a) $\theta/2\theta$ scan showing 50 nm Al (111) and 45 nm $\text{Al}_{0.64}\text{Sc}_{0.36}\text{N}$ (0002) diffraction peaks. **(b)** Rocking curve (ω -scan) of $\text{Al}_{0.64}\text{Sc}_{0.36}\text{N}$ (0002).

Figure R5 (b) shows the rocking curve of the AlScN sample. In **Figure R5 (b)**, the black scatter points represent the experimental data, and the red dashed line indicates the Gaussian fit. The full width at half maximum (FWHM) of the rocking curve is 2.4344° in 45 nm thin AlScN, comparable to values reported in the previous reports⁽¹⁹⁾. This result also confirms that our 45 nm AlScN film has sufficient crystalline quality for device fabrication and testing.

For Piezoresponse force microscopy (PFM) measurements the sample preparation is illustrated in **Figure R6**. In **Figure R6 (a)**, (i) Al/AlScN/Al were deposited in situ on a 6-inch sapphire wafer, with details in the methods section in our paper. (ii) The top Al was wet etched in a dilute HF solution prepared by mixing 5 ml of 2 % HF with 5 ml deionized water, followed by electron beam lithography (EBL). The patterned PMMA resist served as the mask for wet etching, and the pink region denotes the

etched area. (iii) An Au/Ti indicator was patterned by an EBL process and deposited by electron beam evaporation. (iv) Electrical measurements were subsequently conducted to prepare various switched states of the AlScN layer. We applied the voltage on the bottom Al electrode and grounded the top Al. The red box indicates selectively the switched sites of AlScN. (v) We opened only the switched sites by the EBL process and PMMA development. Subsequently, we repeated the dilute HF etch using the same process as in step (ii) to remove the top Al and expose AlScN for PFM test. In practice, PMMA does not form perfectly sharp edges. To prevent residual top Al metal from remaining at the capacitor edge, we left a small margin. As a result, a narrow area next to the switched AlScN region marked in pink was also etched, as depicted in **Figure R6(a)-(v)**. This fact will support later the PFM analysis presented in **Figure R10**. (vi) Consequently, PFM was performed. The PFM analysis was carried out in an Asylum MFP-3D AFM with an Nanosensors PPP-EFM tip of 75 kHz resonant frequency and 2.8 N/m spring constant.

Figure R6 | Preparation of the PFM test sample. (a) Schematics of the sample preparation process. **(b)** Optical microscopy (OM) images of the AlScN PFM sample. The large panel on the left shows the entire array, and the two small panels on the right show magnified views.

In **Figure R6 (b)** the large OM image on the left presents the full patterned array. The red box shows the state after step (v), in **Figure R6 (a)**, used to prepare for the PFM test. The blue box shows the state after step (ii), in **Figure R6 (a)**, used for electrical tests. The right panels provide magnified views labeled “Before” and “After”. “Before” shows the top Al electrode is still present. “After” shows selective removal of the top Al, which exposes AlScN at the intended locations. The indicator is visible in both states and serves as a reference marker that guides us back to the exact switched region and allows precise PFM tip placement.

Figure R7 | PFM Test results of preset $2P_r = 100 \mu\text{C}/\text{cm}^2$. PFM test results for (a) height topography, (b) amplitude and (c) phase images which are prepared for the $100 \mu\text{C}/\text{cm}^2$ of $2P_r$. (d) Enlarged view of the tested region marked in the overview. The highest magnified images with $500 \times 500 \text{ nm}^2$ region of (e) amplitude and (f) phase.

Figure R7 provides the PFM on AlScN that was switched to a partial polarization state with preset $2P_r = 100 \mu\text{C}/\text{cm}^2$. **Figure R7 (a)** shows the AFM topography where partially switched AlScN is present. Particularly, a triangular metal indicator marks the site where pulses were applied. **Figure R7 (b)** and (c) display amplitude and phase, respectively, from the same area. To perform the PFM measurement, an electrical pre-poling process was conducted before imaging. Since our study focuses on polarization defined with respect to negative $2P_r$, the film was first initialized to the M-polar state by applying two positive voltage pulses to the bottom electrode exceeding the coercive voltage. Although a single pulse can align most dipoles, a second pulse was applied to ensure complete switching and stabilization of

the M-polar state. Subsequently, two negative voltage pulses were applied to induce partial polarization. The extent of partial polarization of $100 \mu\text{C}/\text{cm}^2$ was calculated using the conventional PUND method. As discussed in **revised Supplementary Figure S6**, applying two pulses causes two switching events, leading to a larger polarization change than a single-pulse operation. While the targeted switching level was set to $2P_r \approx 100 \mu\text{C}/\text{cm}^2$, the actual polarization change reached approximately $125 \mu\text{C}/\text{cm}^2$, as shown in **Figure R4 (c)**.

The amplitude and phase contrasts in **Figure R7** are not significantly different from those of pristine AlScN, which predominantly exhibits N-polar orientation⁽²⁷⁾. This similarity makes it difficult to distinguish the partially switched region from the pristine area, in **Figure R7(b) and (c)**. Nevertheless, the switched polarization still remains below the saturation level of AlScN. As shown in **Figure 1e**, full polarization in AlScN reaches approximately $200 \mu\text{C}/\text{cm}^2$, confirming that the state observed in **Figure R7** corresponds to partial polarization. However, it is still difficult to observe evidence of partial polarization in **Figure R7(a–c)** because of the relatively large scale.

Figure R7 (d) provides an enlarged view of the selected area, serving as an intermediate link between the overview in **Figure R7 (a)** and the nanoscale scans. **Figures R7 (e) and (f)** display the highest magnification images over a $500 \times 500 \text{ nm}^2$ area, capturing the amplitude and phase, respectively. At this nanoscale resolution, **Figures R7 (e) and (f)** provide clear evidence of partial polarization switching. The amplitude contrast is spatially non-uniform, while the phase shows separate states, confirming that only a fraction of dipoles are reversed within the switched region. Such coexistence of switched and unswitched domains indicates incomplete switching, which are characteristics of partial polarization switching. These features serve as direct nanoscale proof of incomplete switching in the AlScN. The results are consistent with our earlier findings⁽²⁷⁾ and other previous reports^(28, 29), supporting the conclusion that stable intermediate polarization states can be reliably achieved in nitride ferroelectrics, which is theoretically confirmed in a previous report⁽⁶⁾.

Figure R8 presents a comprehensive set of PFM images obtained under different preset $2P_r$ conditions ranging from 10 to $200 \mu\text{C}/\text{cm}^2$. These images were obtained by scanning a wide area ($80 \times 80 \mu\text{m}^2$) in a single acquisition, minimizing errors caused by different measurements. The scan was performed at a very slow rate of 0.1 Hz to further reduce measurement artifacts. In **Figure R8 (a)** show no significant differences between switched and unswitched regions, confirming that the HF etching process was performed uniformly. It also confirms that the applied bias and etch process does not cause significant surface deformation or damage.

In contrast, the amplitude and phase responses shown in **Figure R8 (b)** and **(c)** exhibit a clear dependence on the preset $2P_r$. As discussed in **Figure R7**, the region preset to $2P_r = 100 \mu\text{C}/\text{cm}^2$, which is close to the pristine N-polar predominant AlScN state, shows only subtle contrast differences, making it difficult to distinguish the switched area. The contrast is weakest near $2P_r = 100 \mu\text{C}/\text{cm}^2$ and becomes more pronounced when the preset deviates from this value in either direction. When the preset is increased to $200 \mu\text{C}/\text{cm}^2$, both amplitude and phase images display a pronounced bright region, clearly distinguishing the switched domains from the surrounding matrix. These results confirm that the polarization in AlScN can be continuously adjusted by controlling the preset $2P_r$, allowing fine control over the average polarization density.

Figure R8 | PFM images of AlScN under various preset $2P_r$ conditions. The images consist of (a) topography, (b) amplitude, and (c) phase maps acquired simultaneously in a single scan. The regions indicated by triangular metal indicators correspond to areas where voltage pulses were applied to induce switching. The extent of preset polarization for each region is labeled directly in the images. The red dashed lines in the amplitude and phase images indicate the locations of line profiles, which are analyzed in **Figure R9**.

In **Figure R9**, PFM line profiles were compared under different preset $2P_r$ conditions to highlight the effect of partial polarization. Both amplitude (**Figure R9 (a)**) and phase (**Figure R9 (b)**) profiles clearly distinguish between the switched region (from 5 to 23 μm along the length) and the unswitched region (other regions). Notably, the amplitude and phase increase systematically with larger preset $2P_r$, indicating that the extent of domain reversal scales with the applied polarization. In summary, these observations confirm that the amplitude and phase response reflects the progressive evolution of

partially switched domain fraction with increasing preset $2P_r$. These results support again the nanoscale evidence of controlled ferroelectric in AlScN films.

Figure R9 | PFM amplitude and phase line profiles under various preset $2P_r$ conditions. (a) Amplitude line profiles and (b) phase line profiles are extracted along the red dashed lines indicated in **Figure R8**. Each profile corresponds to different preset $2P_r$ values (10, 50, 100, and 200 $\mu\text{C}/\text{cm}^2$).

In **Figure R10**, we emphasize again the line profile to examine whether HF etching influences the amplitude and phase responses, which could otherwise cause misinterpretation of the ferroelectric signal. All regions A, B, and C in **Figure R10 (a)** share the AlScN surface but differ in their conditions. Region A is electrically switched, region B is unswitched but partially etched by HF due to the fabrication margin left from the EBL process described in step (v) of **Figure R6 (a)**. Region C is unswitched and was completely protected by PMMA during processing, thus remaining close to the as-grown state. This distinction helps us interpret whether surface modification influences the polarization state of AlScN.

As indicated by the white arrow in the inset of **Figure R10 (a)**, the etching process induced margin is clearly visible in region B and is further confirmed in the corresponding height profile. **Figure R10 (d)** shows that the height of region B is ~ 1 nm lower than regions A and C, which remain nearly the same height. While such height topographic variations could in principle influence amplitude and phase detection, our low frequency and optimized measurement conditions eliminate this effect. The amplitude and phase responses reflect only the switching state, independent of the height difference, as verified in **Figure R10 (d)**. This confirms that the observed contrast arises from ferroelectric polarization rather than from etching-related artifacts. In addition, we again confirm the presence of

partial polarization similar to that discussed in **Figure R7**. In **Figure R10**, the pristine region (B and C) is predominantly N-polar, yet a small portion of the domains inherently remain in the M-polar orientation. The switched region (A) also exhibits partial reversal, but with a different fraction of M and N-polar domains. These observations further highlight that partial polarization in AlScN is determined by the relative proportion of M- and N-polar domains.

Figure R10 | PFM images and line profiles across the switched boundary of preset $2P_r = 50 \mu\text{C}/\text{cm}^2$. (a–c) present the PFM topography, amplitude, and phase images. The inset in (a) illustrates the schematic structure of the sample, where arrows indicate the correspondence between the schematic and the PFM images. The red dashed lines mark the positions along which the line profiles were extracted. (d) shows the resulting line profiles of height, amplitude and phase across the boundary, clearly distinguishing the switched and unswitched regions. The red dashed line separates regions A, B, and C. The left part corresponds to region A, the middle to region B, and the right to region C.

Change to the manuscript:

Additional supplementary figures and information have been included in the **revised Supplementary Information** to provide structural analysis by XRD ($\theta/2\theta$ scans, rocking curve) and partial switching of AlScN evidence from PFM imaging and line profiles. **Figure R5** was added as **Supplementary Figure S2**, showing the $\theta/2\theta$ scan and rocking curve of $\text{Al}_{0.64}\text{Sc}_{0.36}\text{N}$ on Al/Sapphire, with the relevant discussion provided in the **revised Manuscript (pages 4, lines 6 to 8)**. **Figures R6–R10** were added as **Supplementary Information S3**, presenting the sample preparation process for PFM and the nanoscale imaging results under various preset $2P_T$ conditions. The relevant discussion is provided in the **revised Manuscript (pages 5, lines 13 to 14)**.

Reviewer's comments:

2.The electrode has important on the endurance for ferroelectric. How about the endurance of AlScN when using other electrode metal, such as, Sc, Cr?

Response:

We thank the reviewer for highlighting the role of electrodes in endurance. We agree that contact metals have a strong influence on switching, leakage, and reliability. In our study, the primary reason for using Al as a top electrode was to preserve the AlScN surface from oxidation. Immediately after AlScN sputtering, Al was deposited in situ to serve as a protective cap. Because AlScN is highly reactive, removal of this Al layer would expose the surface to air, causing rapid oxidation and defect formation that degrade switching and endurance. Retaining the in-situ Al electrode ensures a clean and stable interface, which is critical for reliable device performance. Furthermore, our sputtering system limits the use of alternative metals to maintain AlScN film quality, making Al the most practical and reproducible choice for the top electrode.

Figure R11 | Electrical measurement results of AlScN capacitors with different metal top electrodes. (a) Schematic of the capacitor structure used to evaluate the effect of various top electrode metals (Cr, Ti, Au, and Pd). The top electrode was deposited in situ with a thin Au capping layer to prevent ambient oxidation. (b) Corresponding optical microscopy (OM) image of the fabricated devices. (c - e) Representative DC-IV curves, AC-IV curves, and PUND test results comparing the electrical behavior for each top electrode metal.

We acknowledge that systematic study of different top metals is important for understanding AlScN capacitor behavior. To investigate electrode effects, we fabricated a control set with Cr, Ti, Au, and Pd top electrodes on the same wafer, with devices placed close together to minimize fabrication variations. All electrodes were deposited by sputtering and patterned by electron-beam lithography. A 10 nm Au capping layer was deposited in-situ on Cr, Ti, and Pd electrodes to suppress their native oxidation. **Figure R11** presents the device schematic, optical image, and electrical characteristics. In both DC-IV and AC-IV, higher work function metals exhibit lower current levels, as shown in **Figure R11 (c)** and **R11 (d)**. Furthermore, the ferroelectric switching peak becomes less distinct with deeper work function contacts. In **Figure R11 (e)**, the onset of partial switching shifts toward higher voltages as the work function increases. We anticipate that these behaviors can arise from contact barrier modulation influencing carrier injection. We suggest that low-temperature measurements could help to clarify this interpretation. In addition, the most critical degradation pathway is likely related to native oxide formation once the in-situ Al is removed and the AlScN surface is exposed, leading to interface traps

and accelerated fatigue. At present, we are not aware of prior systematic reports mapping AlScN endurance across different top electrode metals. We therefore believe this topic deserves further experimental investigation in future work, as electrode interfaces and carrier injections play a direct role in endurance.

Figure R12 | Endurance test results of AlScN capacitors with different top metals. All capacitors have a diameter of 20 μm and were tested under a preset $2P_r$ of 200 $\mu\text{C}/\text{cm}^2$. **(a–d)** show the endurance performance with Cr, Ti, Au, and Pd as the top electrode, respectively, arranged in order of increasing work function.

In **Figure R12**, all capacitors have a diameter of 20 μm and were tested at a preset $2P_r$ of 200 $\mu\text{C}/\text{cm}^2$. **Figure R12 (a–d)** corresponds to Cr, Ti, Au, and Pd top electrodes, arranged in order of increasing work function. Here, we observe that endurance decreases markedly with higher work function. This

behavior can be anticipated from **Figure R11**, where deeper work function contacts reduce carrier injection, weaken the switching peak, and require higher operating voltages. To maintain the same $2P_r$, the operating voltage must increase with work function. Notably, in **Figure R12 (c)**, the device initially operated at a relatively low voltage but exhibited severe degradation before the voltage was raised to reach the preset $2P_r$, which indicates that breakdown was accelerated due to the large work function of the top electrode. We confirm that devices with large work function metal as top electrode experience stronger electrical stress and show accelerated breakdown as the work function increases. This agrees with the explanation in the main text that higher operating voltage increases stress and causes faster breakdown.

Figure R13 | Continuous AC-IV hysteresis loops. Seventy consecutive measurements were performed, and the data are presented in intervals of ten cycles for **(a)** Ti and **(b)** Au contacts. This representation allows a clear comparison of the progressive degradation in the electrical response for each electrode condition.

As shown in **Figure R12**, electrodes with deeper work functions display accelerated breakdown. To further probe this effect, continuous AC-IV measurements were conducted over 70 cycles at 10 kHz. In **Figure R13 (a)**, the Ti contact with a relatively shallow work function exhibits a gradual decrease in coercive voltage along with a steady increase in leakage current. However, the distinct switching peak remains clear, confirming ferroelectric switching. By contrast, in **Figure R13 (b)**, the Au contact with a much deeper work function shows decreasing leakage current while the switching peak progressively fades, indicating degradation of the ferroelectric response. The curves were collected in order, starting from green to red.

Although we applied same voltage range to both devices, the Au contact sample degraded faster than the Ti contact case. The deeper work function metal contact might accelerate defect accumulation and charge trapping at the interface between the top electrode and AlScN, which destabilizes domain reversal and shortens the device lifetime. Therefore, the observed differences in AC-IV cycling directly reflect the field-driven degradation process enhanced by high work function contacts.

To the best of our knowledge, no prior reports have systematically investigated the correlation between electrode work function and endurance in AlScN capacitors. Therefore, the observed trend in **Figure R12** highlights an important direction for future study, where deeper work function top contacts and the associated voltage stress should be carefully examined for AlScN ferroelectric devices.

We conclude that Al is the optimized choice of top electrode. Above all, using in-situ Al as the top electrode means that the AlScN surface remains clean, free from native oxide formation which is essential for achieving high-quality devices. Furthermore, our comparative study with metals of different work functions demonstrates that electrodes with lower work functions provide superior endurance. In summary, these results establish Al as the optimized top electrode, ensuring both interface stability and favorable endurance performance.

Change to the manuscript:

In the **revised Manuscript**, we have explained our choice of the in-situ Al top electrode in preference to ex-situ deposited metal contacts. We also emphasize that future studies should systematically investigate different top metal contacts to more clearly elucidate their impact on device performance (**pages 3, lines 8 to 12 in the revised Manuscript**).

Reviewer's comments:

3. The author claims that the endurance is up to 1010 cycle when partial switching is $30 \mu\text{C}/\text{cm}^2$. How about the retention time of partial switching at $30 \mu\text{C}/\text{cm}^2$.

Response:

We thank the reviewer for the insightful comment on an essential retention characteristic of ferroelectric materials. **Figure R14** presents the retention results under various conditions such as 10 kHz with 20 μm diameter, and 1 MHz with 10 μm . Retention was evaluated using a current-based

readout at a low voltage of 6 V, enabling state verification with minimal electrical disturbance. As illustrated in **Figure R14 (a)**, the test protocol consists of two positive RESET pulses followed by two negative SET pulses, with a read step performed after each pair to record the current of the AlScN capacitor.

To reset the dipoles into the M-polar state, 18 V pulses at 10 kHz were applied, which was sufficient to achieve full switching, as confirmed in **Figure 1e**. The conventional $2P_r$ values extracted from the RESET response also indicate complete polarization reversal. These four pulses and subsequent read steps were repeated until the conventional $2P_r$ obtained from the SET pulses reached the target $2P_r$ value. Once the target $2P_r$ was achieved, the device was held for the designated retention time and then read again to evaluate any resistance variation.

Figure R14 | Retention test results under various conditions and device diameters. (a) illustrates the voltage pulse configuration used in the test. (b) and (c) show the retention performance.

The film was first initialized to the M-polar state. We then induced partial switching to the N-polar state to preset $2P_r$ values of 10, 50, and 100 μC/cm². Because ferroelectric conduction is polarization dependent, the current after RESET is larger and defines the low resistance state (LRS), whereas the

current after SET is smaller and defines the high resistance state (HRS). To exclude any possible reading influence, RESET and SET are alternated again and the applied voltage is tuned until the $2P_r$ computed from the SET pulse returns to the target value.

We define I_{on} as the current read after RESET, I_{off_1} as the current read after the first SET, and I_{off_2} as the current read after the retention interval. Retention is quantified by comparing I_{on}/I_{off_1} and I_{on}/I_{off_2} . If strong depolarization occurred during the wait time, the ratio (I_{on}/I_{off_2}) would approach one. As shown in **Figure R14 (b) and (c)**, the ratio shows no meaningful degradation. It remains nearly constant up to ten thousand seconds, with a slight increase at most. This small increase is likely due to partial switching of residual M-polar regions toward the N-polar direction, which can be attributed to the preferential imprinting of the inherently N-polar state in pristine AlScN.

These observations are consistent with previous reports demonstrating robust retention in AlScN-based capacitors, both under full polarization switching conditions^(30 - 32), and in multistate FeRAM configurations^(7, 33). Our findings further confirm that AlScN maintains stable $2P_r$ even in partially switched states across various frequencies and device sizes.

Change to the manuscript:

In revised Supplementary Information, **Figure R14** was added as **Supplementary Figure S12**, presenting the retention characteristics of partial switching in the AlScN capacitor. The relevant discussion is provided in the **revised Manuscript (pages 10, lines 25 to 32)**.

Reviewer's comments:

4. In figure.1f, the conventional $2P_r$ is $50\mu C/cm^2$, $100\mu C/cm^2$ and $200\mu C/cm^2$, the endurance increase by increase frequency. When conventional $2P_r$ is $10\mu C/cm^2$, the endurance has no relationship with the frequency. Why?

Response:

We thank the reviewer for this thoughtful question. In **Figure 1f**, in our paper, the conventional $2P_r$ is maintained at a constant value during the endurance test by a self-voltage adjustment algorithm, as discussed in our work. As frequency increases, the coercive voltage also rises, which forces the device to operate at higher voltages. This means that to achieve the same amount of partial polarization

switching, a larger voltage is required when the frequency increases. According to well-known switching models, such as Kolmogorov–Avrami–Ishibashi (KAI) and nucleation-limited switching (NLS), the switching time depends on the applied voltage^(34, 35). At higher frequencies, the available pulse duration becomes shorter. This limited time window means that switching must occur more quickly. Since the switching dynamics of dipoles are governed by the operating frequency, a shorter pulse width effectively constrains the available time window for dipole switching. To achieve the same amount of polarization switching within this shorter interval, the applied voltage must be increased to accelerate the switching kinetics. Therefore, the algorithm compensates by raising the applied voltage incrementally to maintain the same level of polarization switching. Unfortunately, this results in larger leakage current and a stronger electric field across the AlScN layer, thereby accelerating leakage-driven degradation and promoting earlier breakdown phenomena. At higher preset $2P_r$ values, this stress factor becomes the dominant mechanism determining the overall frequency dependence. As the preset $2P_r$ increases, higher operating voltages are required, which in turn accelerates dielectric breakdown. Consequently, the relationship between frequency and endurance is positive when $2P_r$ is 50, 100, or 200 $\mu\text{C}/\text{cm}^2$, as shown in **Supplementary Figure S6**, and is further supported by the correlation between breakdown and coercive electric field in **Supplementary Figure S7**.

At the same time, however, the rise in frequency has a second effect that partially counterbalances the stress increase. Shorter pulse width at higher frequencies effectively reduces the time that the device is exposed to strong electric stress, which enhances the tolerance against breakdown. In the small partial-switching case, the required absolute voltage is relatively small and the corresponding leakage currents are much lower than in full-switching or higher partial-switching conditions. These factors lower the per-cycle electrical stress and lead to a distinct frequency dependence of endurance in the low-partial-switching regime ($2P_r \approx 10 \mu\text{C}/\text{cm}^2$). This observation clarifies why our devices operated in the low partial polarization regime show a markedly opposite frequency trend compared to those operated at high preset $2P_r$ values.

In conclusion, two opposing mechanisms govern endurance as a function of frequency. The first is the increase in coercive voltage that enforces higher operating voltage and accelerates degradation. The second is the shortening of pulse width, which mitigates field stress and can extend endurance. The balance between these effects depends on the target polarization level. At 50–200 $\mu\text{C}/\text{cm}^2$, the higher voltage stress dominates even though pulses shorten, leading to significant degradation. In contrast, at 10 $\mu\text{C}/\text{cm}^2$, the absolute operating voltage is relatively low and the pulse width is reduced. Therefore,

the decreased stress exposure time prevents severe degradation and results in a frequency trend opposite to that at higher preset values. This comprehensive interpretation explains the difference of trends and aligns with the reorganized plots in **revised Supplementary Figure S8** as well as the breakdown–coercive field relationship shown in **revised Supplementary Figure S9**. We believe that this expanded explanation provides a clearer understanding of why endurance decreases with frequency at larger $2P_r$ values, while at smaller $2P_r$ values the trend is opposite and endurance increases with frequency.

Change to the manuscript:

In the **revised Manuscript**, we have expanded our explanation of the background understanding regarding the distinct endurance trends as a function of preset $2P_r$ (**pages 7, lines 7 to 11**). In the **revised Supplementary Information**, we further emphasize that additional theoretical studies are required to investigate the relationship between electrical stress and endurance with **revised Supplementary Figure S8**.

Reviewer’s comments:

5. Is this method general for different thickness AlScN?

Response:

We thank the reviewer for this important question and emphasize that such comments are crucial for advancing the field. At present, we are investigating thinner AlScN films in a follow-up study. We do not yet observe a consistent trend of endurance under varying thicknesses and conditions. For this reason, we prefer not to speculate beyond the available evidence.

However, prior reports in our group motivate this direction. Ultrathin AlScN ferroelectric have been demonstrated with interlayer engineering, enabling scaling down to 5–20 nm with reduced switching voltages^(7, 33). Similar behavior was also observed without interlayers, where direct scaling preserved ferroelectric switching current response⁽¹⁹⁾. Capacitors with thickness approaching 5 nm show reliable switching. Scaled devices at 5 and 10 nm also exhibit low-voltage operation and stable multistate performance⁽⁷⁾. Together, these studies confirm that ferroelectric partial switching can persist across an ultrathin regime.

There are several physical factors about endurance to consider when the thickness decreases. As we discussed in our paper, the ratio between breakdown field and coercive field is a key parameter that helps to predict endurance. Both breakdown and coercive fields increase as the thickness is reduced^(7, 36). Whether endurance improves or degrades depends on which parameter increases more rapidly. If the breakdown field rises faster than the coercive field, endurance may improve. However, if the coercive field dominates, endurance is expected to degrade. In addition, leakage current and series resistance vary with thickness. Although their exact effects are not yet known, we suspect that they have a significant impact on endurance. Moreover, interface contributions also grow as films become thinner, and incomplete screening enhances the depolarization field^(36, 37). Overall, these combined effects make it hard to anticipate how endurance will evolve with AlScN thickness.

In conclusion, while prior studies confirm that ferroelectric partial switching persists in the ultrathin regime, endurance behavior remains an open question. Our ongoing systematic study will clarify these dependencies and provide quantitative results in the future.

Change to the manuscript:

In the **revised Manuscript**, we have added a new sentence to address the reviewer's question on thickness dependence. The relevant discussion notes that robust ferroelectric current response is still observed even at ultrathin dimensions, while also indicating that both the coercive field and the breakdown field vary with thickness, which are critical parameters for endurance and reliability (**pages 15, lines 5 to 12**).

Reviewer's comments:

6.How many devices was conducted in this paper?

Response:

Each condition was tested in few endurance measurements, as shown in **Figure R16**. While additional data would have allowed more detailed statistical analysis, each endurance test required several hours to as long as a week, making a larger dataset impractical. Despite this limitation, the results consistently revealed reproducible behavior under identical measurement conditions.

In particular, the voltage trajectories required to maintain the preset $2P_r$ were nearly identical in all

repeated tests, demonstrating the reliability of our optimized protocol. These reproducible results confirm that endurance is governed primarily by the intrinsic fatigue dynamics of AlScN and the voltage adjustment algorithm, rather than by stochastic variations. Representative conditions are selectively presented in **Figure R16** to illustrate the essential trends across the full dataset.

Figure R16 | **Additional endurance test data under various conditions.** (a–e) show the results for 10 μm diameter capacitors measured at 100 kHz, where the conventional $2P_r$ is maintained at 100 $\mu\text{C}/\text{cm}^2$. (f–j) present the results for 40 μm diameter capacitors measured at 1 MHz, with the conventional $2P_r$ maintained at 50 $\mu\text{C}/\text{cm}^2$. (k–o) display the results for 20 μm diameter capacitors measured at 1 MHz, with the conventional $2P_r$ maintained at 10 $\mu\text{C}/\text{cm}^2$.

Change to the manuscript:

In the **revised Manuscript**, we have added sentences describing the practical challenges in performing a similar number of endurance tests across all devices and conditions for statistical analysis (**pages 6, lines 2 to 4**). To address this, we selectively added representative data as **Supplementary Figure S5 (Figure R16)**, which illustrates that under identical test conditions the voltage evolution during endurance cycling follows nearly identical trajectories, highlighting the reproducibility of the results and the overall endurance trends.

Reviewer’s comments:

7. In this article, the fatigue mechanism should be discussed detailly?

Response:

We thank the reviewer for raising the question on the fatigue mechanism. In our work we monitor fatigue through the current response and through the evolution of the operating voltage. Revised **Supplementary Information S5** summarizes both the fixed voltage and the fixed $2P_r$ endurance. **Revised Supplementary Figure S5-2** presents the phase evolution under endurance and directly supports the mechanism discussed below.

In the fixed voltage case, we track the N and D peaks. N reflects ferroelectric switching while D captures non switching current which has a high leakage contribution. At low voltage the device shows a wake-up phase. Only the N peak grows as internal bias relaxes and pinned domains become switchable^(11, 38 – 42), while leakage remains suppressed. As cycling proceeds, leakage rises and both N and D increase. In this regime, interfacial traps generate a relaxation tail in the current response, primarily arising from Maxwell–Wagner relaxation and Curie–von Schweidler relaxation. Most of this tail decays during the N pulse, but a residual portion persists into the D pulse. As a result, both N and

D peaks increase. This unequal non-switching contributions in N and D cause the conventional $2P_r$ to be overestimated. To mitigate this trap-induced artifact, reducing the applied voltage is necessary, which helps suppress trap activity and minimizes the relaxation tail^(8, 9, 25, 42, 44). With the onset of fatigue, the N peak decreases because charged defects pin domains⁽⁴⁵⁾ and a non-ferroelectric interfacial layer grows which reduces the switched volume and raises the effective switching barrier⁽⁴⁶⁾. In the case of leakage current, the D peak clearly reveals the behavior. At first, vacancies enhance leakage along grain boundaries, causing the D peak to grow. With continued electrical stress, the current paths gradually collapse or stabilize, leading the D peak to decline or level off^(42, 46, 47). These trends match **revised Supplementary Figure S5-2** where D reaches a maximum and both N and D subsequently fall. We attribute the N decrease to interfacial non ferroelectric layer thickening and defect pinning reported for AlScN and the D evolution to vacancy percolation and oxygen assisted structural degradation observed under cycling.

In the fixed $2P_r$ case the algorithm adjusts the pulse amplitude to maintain the preset value. During early wake up the required voltage decreases as more domains participate in switching. With further cycling, leakage grows and both N and D increase for the same reasons as above. Once fatigue starts the system requires a higher voltage to sustain the target $2P_r$, and the turning point from decreasing to increasing voltage is a robust marker of fatigue onset. **Revised Supplementary Figure S5-4a** and **S5-4b**, which are organized in parallel with the fixed-voltage analysis, clearly illustrate this evolution. By aligning the adaptive voltage trajectories in **revised Supplementary Figure S5-4** with the phase map in **revised Supplementary Figures** from **S5-1** to **S5-3**, one can trace the same sequence of wake-up, leakage growth, fatigue onset, and eventual breakdown of AlScN capacitors. We observe this qualitative trend consistently across the tested frequencies, diameters, and preset $2P_r$ values. At a small preset such as $10 \mu\text{C}/\text{cm}^2$ the current response remains stable for very long cycling. In this regime the evolution of the applied voltage is barely visible, confirming that the electrical stress is strongly suppressed.

We fully acknowledge the reviewer's comment. Our focus was largely on the methodological and engineering advances that improved endurance significantly, with less attention given to the fundamental fatigue mechanisms. In this revision, we have expanded the discussion by incorporating a detailed explanation of the fatigue processes as outlined above, thereby providing a more balanced scientific perspective on our work. While we recognize that this area still requires further investigation, we plan to pursue direct imaging studies such as PFM or TEM in follow-up work to strengthen the mechanistic understanding.

Change to the manuscript:

In the **revised Supplementary Information S5**, additional information illustrates the phase evolution and adaptive voltage trajectories during endurance cycling (**pages 50, lines 4 to 23**).

Reviewer #3 (Remarks to the Author):

This study demonstrates an impressive breakthrough in enhancing the write cycling endurance of wurtzite ferroelectric AlScN to over 10^{10} cycles via controlled partial polarization switching and device scaling, representing a thousand-fold improvement over previous benchmarks. The research addresses a critical challenge in ferroelectric memory applications, with well-designed experiments and comprehensive data supporting the findings. While the work is innovative and impactful, several aspects of the mechanism interpretation, comparative analysis, and experimental details could be further refined.

Response:

We sincerely thank the reviewer for recognizing the endurance breakthrough and its relevance to ferroelectric memory. The comment highlights the value of the improvement while also suggesting that interpretation, comparison, and methodology could be refined. We deeply appreciate these constructive perspectives.

In response, we have revised the manuscript with additional clarifications in the discussion, a more explicit comparative context, and improved descriptions of the experimental procedures. We hope that these revisions enhance both the clarity and the reproducibility of the work. In the following, we provide point-by-point responses to address the reviewer's specific comments.

Reviewer's comments:

1. The study attributes enhanced endurance to "incomplete domain switching" during partial polarization, but lacks microscopic evidence (e.g., TEM/PFM images) of domain structure evolution. A theoretical model explaining domain wall pinning or defect-mediated switching in AlScN would strengthen the mechanism interpretation.

Response:

We thank the reviewer for raising this important point. Our work demonstrates that enhanced endurance in AlScN capacitors originates from controlled partial switching, where only a fraction of domains switched polarization. We have included detailed Piezoresponse Force Microscopy (PFM) analysis in **Figure R18**. The preparation process for PFM measurements is outlined in **Figure R17**.

Al/AlScN/Al capacitors were fabricated in situ on a 6-inch sapphire wafer. The top Al was partially removed by selective wet etching through electron-beam lithography (EBL)-patterned PMMA masks, thereby exposing predefined regions of the AlScN layer for PFM testing. An Au/Ti indicator structure was patterned as a positional reference to ensure precise alignment of the PFM probe with the previously switched regions. Prior to PFM, the capacitors were electrically biased using the partial polarization algorithm described in our manuscript to establish intermediate switched states.

Figure R17 | Preparation of the PFM test sample. (a) Schematics of the sample preparation process. (i) Al/AlScN/Al layers deposited in situ on a 6-inch sapphire wafer, (ii) top Al wet etched in dilute HF (5 ml of 2 % HF + 5 ml DI water) using EBL-defined PMMA mask, leaving the etched area in pink, (iii) Au/Ti indicator patterned by EBL and deposited by e-beam evaporation, (iv) electrical switching of AlScN using the partial polarization algorithm, red box marking the switched sites, (v) selective re-etching through EBL openings to remove top Al and expose switched AlScN, with a narrow etched margin due to imperfect PMMA coverage, and (vi) PFM performed using an Asylum MFP-3D AFM with a PPP-EFM tip (75 kHz resonance, 2.8 N/m spring constant). (b) OM image of the patterned array: red box shows the state after step (v) for PFM measurement and blue box shows the state after step (ii) for electrical tests, and magnified panels illustrate “Before” (top Al

remains) and “After” (Al removed, AlScN exposed).

As shown in **Figure R18**, the AFM topography highlights the exposed AlScN region, and a triangular metal indicator denotes the location where electrical pulses were applied. Following the pre-poling electrical procedure, the film was first initialized to the M-polar state by applying two positive pulses exceeding the coercive voltage, and partial polarization was then induced by two negative pulses. The targeted switching level was set to $2P_r \approx 100 \mu\text{C}/\text{cm}^2$ based on conventional PUND analysis, although the actual polarization change can slightly exceed this value ($\approx 125 \mu\text{C}/\text{cm}^2$) due to two-step switching, as discussed in **revised Supplementary Information S6**.

Figure R18 | PFM test results of preset $2P_r = 100 \mu\text{C}/\text{cm}^2$. PFM test results for (a) height topography, (b) amplitude and (c) phase images which are prepared for the $100 \mu\text{C}/\text{cm}^2$ of $2P_r$. (d) is the magnified image to show the region where PFM was conducted. The most magnified images with $500 \times 500 \text{ nm}^2$ region of (e) amplitude and (f) phase.

The PFM amplitude and phase images in **Figures R18 (b) and (c)** show that the overall contrast across the switched region is not strongly distinguished from pristine AlScN, which is predominantly N-polar in the as-grown state. This similarity makes the partially switched region difficult to identify at the large imaging scale. To bridge this gap, **Figure R18 (d)** presents an intermediate magnification view, connecting the overview with the nanoscale scans.

At the highest magnification (**Figures R18 (e) and (f)**, $500 \times 500 \text{ nm}^2$), the signature of incomplete switching becomes clear. The amplitude signal is spatially non-uniform and the phase exhibits distinct but mixed polarization orientations. The coexistence of N and M-polar states within the same nanoscale region confirms that the polarization has not reached the saturated state of AlScN ($\approx 200 \text{ } \mu\text{C}/\text{cm}^2$), as shown in **Figure 1e**, and instead resides in a stable intermediate configuration. Such spatially mixed domain configurations are a direct nanoscale indicator of partial polarization switching. The results are consistent with our earlier finding⁽²⁷⁾ and other previous group reports^(28, 29), supporting the conclusion that stable intermediate polarization states can be reliably achieved in nitride ferroelectrics.

To complement these experimental results, we further introduce a theoretical framework describing incomplete domain switching in AlScN. Recent studies on wurtzite ferroelectrics have established that at higher Sc content, the switching pathway transitions from a collective cation-anion sublattice displacement to a sequential inversion of tetrahedral units via intermediate nonpolar structures^(6, 48, 49). This mechanism lowers the effective switching barrier and enables the stabilization of metastable intermediate polarization states. As a result, this theoretical background demonstrates that partial polarization is theoretically feasible. These intermediate states can remain stable over repeated cycling, allowing the system to sustain non-fully switched configurations without relaxation. Under such partial polarization, the effective electric field acting on each domain is reduced, suppressing the nucleation rate of reverse domains. Consequently, the per-area probability that a critical nucleus will develop, and collapse diminishes, lowering the instantaneous hazard for breakdown and extending the time to failure⁽⁸⁷⁾. Our electrical data, showing extended endurance beyond 10^{10} cycles under partial polarization ($2P_r \approx 10 - 100 \text{ } \mu\text{C}/\text{cm}^2$) compared to $\sim 10^8$ cycles for full polarization reversal ($2P_r \approx 200 \text{ } \mu\text{C}/\text{cm}^2$), are consistent with this theoretical model.

The observed PFM features are consistent with a nucleation limited and defect pinned switching regime in wurtzite AlScN⁽⁵⁰⁻⁵³⁾. Under electric fields below the coercive voltage, polarization switching proceeds by nucleation at easy sites and short-range domain wall propagation rather than a collective 180° flip^(54, 55). When a wall encounters a pinning center, bound charge and the associated internal field accumulate at the local wall segment. At a charged wall segment the internal field points opposite to the external drive. The effective driving field on the wall is therefore reduced. Extra work is then required to propagate the wall. In our films ($\sim 50 \text{ nm}$) incomplete screening at the electrodes and at interfacial layers further strengthens this effect^(41, 56-62). These results show a metastable domain pattern that explains the mixed amplitude contrast and the distinct phase response in **Figure R18**. The phase

images show pinned domains and limited wall motion in partially switched AlScN, which persist until higher fields are applied.

In summary, our microscopic PFM measurements and theoretical support provide a framework for understanding enhanced endurance in AlScN. The stabilization of partial polarization states by domain wall pinning and intermediate structural states help the material to avoid the extreme electric field stress required for full switching. Consequently, fatigue and electrical stress are mitigated, Joule heating is reduced, and the overall reliability of the capacitor is significantly improved. These insights not only strengthen our interpretation but also highlight a general principle by which partial polarization control can be used to extend the operational lifetime of wurtzite AlScN ferroelectrics.

Change to the manuscript:

In the **revised Supplementary Information**, Figures **R17** through **R18** have been included as **revised Supplementary Information S3**, illustrating the sample preparation procedure for PFM along with nanoscale imaging results obtained under different preset $2P_r$ conditions. The relevant discussion is provided in the **revised Manuscript (pages 5, lines 13–14)**.

Reviewer's comments:

2. The benchmarking in Fig. 4 omits recent advancements in 2D ferroelectrics (e.g., CuInP₂S₆) and other nitrides (e.g., ScGaN). Including these would more robustly validate AlScN's performance edge.

Response:

We thank the reviewer for pointing out the importance of benchmarking against other emerging ferroelectrics. In fact, our benchmarking in **Figure 4**, which is in our main paper, already includes representative data from both CuInP₂S₆ (CIPS) and ScGaN. We also attempted to collect additional recent reports that explicitly provide both $2P_r$ values and endurance test results for these materials. However, to the best of our knowledge, such data are rarely reported together, making a consistent comparison difficult. This reflects a broader challenge in the field, where many studies on emerging 2D ferroelectrics or alternative nitrides emphasize switching phenomena, but do not yet provide systematic endurance data.

Figure R19 | Benchmarking of ferroelectric endurance performance. A systematic comparison of AlScN endurance characteristics with previously reported ferroelectric materials. The red markers represent data from this study. Among the multiple data points collected under identical x-axis conditions, the most representative dataset was selected. Benchmarking maximum endurance cycles against effective $2P_r$. This figure is presented as **Figure 4a** in our main paper.

We would be grateful if the reviewer could recommend specific recent publications that report both polarization and endurance for additional systems. We will gladly incorporate such data in the revised benchmarking plot.

Reviewer’s comments:

3. Quantitative comparison of energy consumption per cycle between AlScN and HfO₂-based ferroelectrics is missing, which is critical for memory applications. Discussing this trade-off would enhance practical relevance.

Response:

We thank the reviewer for highlighting the importance of quantitatively comparing the energy consumption per cycle between AlScN and HfO₂-based ferroelectrics from the perspective of memory

applications. This is indeed an essential point for evaluating practical significance.

Table R1 | Energy Consumption per Cycle for AlScN vs HfO₂ Ferroelectrics

Material	Thickness & Diameter [nm & μm]	2P _r [μC/cm ²]	Frequency	Energy consumption				Ref.
				E _{PN} [nJ]	E _{UD} [nJ]	E _{PN} / (2P _r × Area) [mJ/μC]	E _{UD} / (2P _r × Area) [mJ/μC]	
Al _{0.67} Sc _{0.36} N	45 & 10	200	10 kHz	23.19	16.7	0.1477	0.1063	Our works
Al _{0.67} Sc _{0.36} N	45 & 10	100	10 kHz	6.67	9.9	0.0849	0.1260	
Al _{0.67} Sc _{0.36} N	45 & 10	50	10 kHz	4.02	6.16	0.1022	0.1568	
Al _{0.67} Sc _{0.36} N	45 & 10	10	10 kHz	0.16	0.07	0.0200	0.0090	
Al _{0.67} Sc _{0.36} N	45 & 10	50	100 kHz	3.84	4.04	0.0978	0.1030	
Al _{0.67} Sc _{0.36} N	45 & 10	50	1 MHz	5.12	2.56	0.1302	0.0652	
Al _{0.67} Sc _{0.36} N	45 & 40	50	10 kHz	46.32	78.5	0.0738	0.1250	
Al _{0.72} Sc _{0.28} N	5 & 10	100	1 MHz	12.57	10.29	0.1600	0.1310	19
Al _{0.72} Sc _{0.28} N	5 & 10	250	1 MHz	25.27	19.29	0.1287	0.0983	19
Al _{0.72} Sc _{0.28} N	27 & 25	170	1 MHz	406.80	272.78	0.4875	0.3269	19
Al _{0.72} Sc _{0.28} N	18 & 25	200	1 MHz	478.15	397.97	0.4870	0.4054	19
Al _{0.72} Sc _{0.28} N	10 & 10	200	1 MHz	51.92	47.03	0.3305	0.2994	19
Al _{0.68} Sc _{0.32} N	10 & 10	200	1 MHz	68.92	57.39	0.4388	0.3654	63
Al _{0.7} Sc _{0.3} N	10 & 1.5	225	1 MHz	0.93	0.73	0.2344	0.1844	64
Al _{0.7} Sc _{0.3} N	40 & 110	175	500 kHz	877.72	819.35	0.0528	0.0493	42
Al _{0.7} Sc _{0.3} N	40 & 70	140	500 kHz	584.67	207.68	0.1085	0.0386	45
Hf _{0.5} Zr _{0.5} O ₂	10 & 226	36	1 kHz	98.52	18.71	0.0069	0.0014	65
Hf _{0.5} Zr _{0.5} O ₂	10 & 113	60	2 kHz	31.8	~ 0	0.0053	~ 0	66
Hf _{0.5} Zr _{0.5} O ₂	1.5 & 56	13	500 kHz	2.28	2.1	0.0069	0.0062	67
Hf _{0.5} Zr _{0.5} O ₂	12.3 & 40	16.4	1 MHz	4	0.76	0.0195	0.0037	68
Zr _{0.33} Hf _{0.67} O ₂	10 & 1.35	10	1 kHz	0.03	0.002	0.1790	0.0140	69
ZrO ₂	10 & 113	51	1 kHz	111.22	30.4	0.0218	0.0059	69

To quantify energy consumption, we integrated ($\int V(t) \cdot I(t) dt$) over each pulse window in PUND measurements, from the onset to the end of the applied voltage. In practice, the energy stored in the device and the energy dissipated during operation should be distinguished. However, due to the lack of detailed material parameters reported in prior studies, such separation could not be made in **Table R1**. Nevertheless, we note that this distinction was considered in our followed analysis and interpretation. The four pulse energies are denoted as E_P, E_U, E_N, and E_D. We define E_{PN} = E_P + E_N, corresponding to switching windows, and E_{UD} = E_U + E_D, corresponding to non-switching windows. While E_{UD} may still include a small extent of switching contributions under partial polarization switching, it gives an upper-bound estimate of non-switching losses. To prepare the comparison in **Table R1**, we collected data

from previous reports^(19,42,45,63-69). As in **revised Supplementary Figure S6** where frequency was calculated from pulse configuration parameters, we also extracted pulse information from prior reports and recalculated the corresponding frequencies using our method to ensure consistency in **Table R1**.

In **Table R1**, systematic trends were confirmed across all cases. The results show that power consumption increases steadily with $2P_r$. For example, at 45 nm thickness and 10 μm diameter, E_{PN} rises from 0.16 to 23.19 nJ as $2P_r$ increases from 10 to 200 $\mu\text{C}/\text{cm}^2$. E_{UD} also increases with $2P_r$ due to higher applied voltages and leakage contributions.

AlScN tends to show higher energy consumption than HfO₂ based ferroelectric material under similar test conditions. AlScN inherently has a larger $2P_r$, which means more energy is stored in the device. Since the stored energy is also counted as part of the consumed energy in this calculation, the evaluated energy consumption inevitably appears larger. However, in the case of small $2P_r$, the energy consumption of AlScN can approach that of HfO₂-based ferroelectric devices. For instance, in the 45 nm thickness and 10 μm diameter AlScN case at 10 kHz with partial $2P_r = 10 \mu\text{C}/\text{cm}^2$, the energy consumption values are close to those reported for HfO₂, showing that partial switching can yield comparable energy use. This highlights that while absolute energy use is generally larger for AlScN, partial polarization operation can mitigate the difference and enable competitive performance.

The higher energy consumption of AlScN compared to HfO₂-based ferroelectrics arises from its intrinsically larger remanent polarization values and the higher coercive fields required to switch them. Larger polarization leads to higher switching currents, and higher coercive fields necessitate higher applied voltages. To enable a more meaningful comparison between the two material systems, we normalized the switching energy by $2P_r$, yielding the parameter $E_{PN} / (2P_r \times \text{Area})$. This metric mathematically and physically is same as the effective voltage experienced during unit polarization switching.

Interestingly, the calculated effective voltage is often significantly higher than the externally applied pulse amplitude. This indicates that a portion of the supplied electrical potential energy on the AlScN is dissipated through non-switching processes such as leakage current, defect charging, or thermal losses, rather than being fully converted into ferroelectric switching work. In contrast, for an ideal ferroelectric with negligible leakage, $E_{PN} / (2P_r \times \text{Area})$ would mathematically converge to the applied voltage itself. Therefore, the observed deviation from the applied voltage indicates that additional energy is dissipated through non-ideal processes inherent to real devices.

Notably, our results reveal that the effective voltage markedly decreases when partial polarization switching of $10 \mu\text{C}/\text{cm}^2$ is applied. This observation suggests that the energy consumed per unit polarization is more efficiently utilized for actual dipole switching rather than dissipated through parasitic loss channels. In other words, under limited switching conditions, the supplied energy is concentrated on ferroelectric switching, leading to minimal additional material degradation. This behavior demonstrates that partial polarization operations not only reduce the effective energy cost but also promotes more stable and less destructive switching dynamics, offering a promising route toward energy-efficient and reliable AlScN-based ferroelectric devices.

However, the ideal minimum energy consumption per unit polarization is still limited by the coercive voltage itself. In other words, minimizing the effective energy per unit polarization requires reducing the coercive voltage. This can be achieved by scaling down the ferroelectric thickness. The relatively large AlScN thickness used in this study contributes to the high coercive voltage. Consequently, the elevated effective switching energy observed here. Previous our reports on thinner AlScN films demonstrate that scaling reduces the coercive voltage⁽¹⁹⁾, offering a clear and practical pathway toward more energy-efficient ferroelectric operation. Such optimization directly enhances switching efficiency and will be crucial for achieving energy-competitive ferroelectric operation in future AlScN-based memory technologies.

In addition, the contribution from parasitic capacitance and resistance should also be considered. These parasitic elements can induce additional energy dissipation that does not contribute to ferroelectric switching. Because their absolute values remain nearly constant regardless of device area, their relative influence becomes more pronounced in smaller capacitors. Consequently, the apparent energy consumption per unit area appears larger for smaller devices. As the device size scales down, careful engineering of parasitic capacitance and resistance becomes increasingly important. This trend is observed not only in AlScN but also in HfO₂-based ferroelectrics, as summarized in **Table R1**.

In summary, our recalculated comparison indicates that AlScN generally consumes more energy per cycle than HfO₂-based ferroelectrics. At the same time, the observed variations with frequency, electrode diameter, and partial $2P_r$ show that energy use can be modulated through careful design. These results point to the need for continued engineering optimization to balance endurance, polarization density, and energy efficiency in future memory applications.

Change to the manuscript:

In the **revised manuscript**, we added a relevant discussion that describes the quantitative comparison of energy consumption per cycle between AlScN and HfO₂-based ferroelectrics (**pages 15, lines 18–22**). To explain this in detail, we introduced **Table R1**, which systematically summarizes the recalculated energy consumption values from our work and prior reports using a consistent PUND-based framework. For clarity, this table is provided in the **revised Supplementary Information** as **Supplementary Table S1**. This addition provides a direct benchmark between AlScN and HfO₂ devices and highlights the role of partial switching in reducing energy consumption.

Reviewer's comments:

4. The AlScN deposition parameters (e.g., sputtering pressure, target purity) in the Methods section are insufficient for reproducibility. Specify details such as chamber pressure, N₂/Ar flow ratio, and target material specifications.

Response:

We thank the reviewer for pointing out the importance of providing detailed deposition parameters to ensure reproducibility. In the **revised manuscript**, we have now expanded the Methods section to include the full growth conditions of AlScN films. Specifically, the films were deposited in an Evatec CLUSTERLINE 200 II pulsed DC sputtering system using high-purity Al (99.999%) and Sc (99.99%) 100 mm diameter targets, powered at 900 W and 700 W, respectively. The chamber base pressure was maintained at 4×10^{-8} mbar. During deposition, nitrogen was supplied at a constant flow of 30 sccm without any Ar gas. The substrate temperature was held at 350 °C throughout growth. In addition, an in-situ Al bottom electrode was employed to provide a suitable (111)-oriented template for c-axis growth of AlScN, as commonly reported in the literature. These details have been incorporated into the Methods section.

Change to the manuscript:

In the **revised manuscript**, detailed deposition parameters of AlScN films have been added to the Methods section to ensure reproducibility (**pages 16, lines 8-13**).

Reviewer's comments:

5. While the study mentions defect density reduction with electrode scaling, a quantitative statistical model (e.g., Weibull distribution) for breakdown probability across different sizes would strengthen the theoretical framework.

Response:

Figure R20 | Dependence of electrode diameter on breakdown and coercive fields. (a) Relationship between electrode diameter and ferroelectric parameters including coercive field (E_C , bottom panel), breakdown field (E_{BD}), and their ratio (E_{BD}/E_C , top panel). (b) Weibull probability plots of breakdown field for different diameters. And (c, d) reorganized the Weibull plot and corresponding linearization to visualize the statistical reliability of MFM Breakdown voltage across different sizes. Symbols represent experimental data, and the dashed line indicates the fitted plots.

We thank the reviewer for highlighting the importance of a quantitative statistical framework. In the revised manuscript, we now include a Weibull analysis of breakdown probability as a function of electrode size. The extracted distributions follow the classical weakest-link model, where breakdown probability scales inversely with electrode area⁽⁷⁰⁻⁷²⁾. Importantly, the Weibull parameter (β), obtained

from the slope of the probability distribution plots, is remarkably high. A large β is consistent with fracture and dielectric breakdown statistics, signifying a uniform intrinsic failure mechanism rather than random extrinsic defects. Furthermore, the probability plots reveal a systematic shift of the breakdown field toward higher values as electrode diameter decreases, directly reflecting the lower chance of defect initiation in smaller capacitors. This scaling trend is consistent with recent reports on nitride ferroelectrics where uniformity improves with reduced device area⁽³³⁾. Overall, the incorporation of Weibull statistics provides a rigorous and widely accepted framework that strengthens our conclusion. Top electrode scaling effectively suppresses breakdown events.

In summary, reducing the electrode diameter not only suppresses breakdown events but also improves endurance, allowing devices to survive significantly longer.

Change to the manuscript:

In the **revised Manuscript**, we added a short description of the Weibull statistical analysis for breakdown probability (**pages 11, lines 12–14**). The corresponding results are provided in the **revised Supplementary Information as Supplementary Figure S14**. These plots highlight the weakest-link scaling and the consistently high β values. They also show a systematic shift of the breakdown field with varying electrode size.

Reviewer's comments:

6. The effect of partial polarization on read/write speed and retention property, which are important practical metrics, were not discussed. Addressing these via additional tests or theoretical predictions would be valuable.

Response:

We thank the reviewer for raising this important point regarding the effect of partial polarization on read/write speed and retention properties. To address this, we conducted systematic retention measurements under different operating frequencies, as summarized in **Figure R21**. The results show that the retention characteristics of AlScN capacitors remain essentially unaffected when the cycling frequency is varied from 10 kHz to 1 MHz. This observation clearly indicates that high-speed operation does not induce any measurable degradation of stored polarization states in our devices. And these retention results are consistent with the previous reports^(7, 33).

Figure R21 | Retention test results under various conditions and device diameters. (a) illustrates the voltage pulse configuration used in the test. (b) and (c) show the retention performance.

These findings suggest that the partial polarization state of AlScN is highly stable against depolarization or relaxation effects even under rapid cycling. AlScN ensures that once domains are switched, they remain pinned, while the intrinsically low dielectric permittivity suppresses depolarization-driven relaxation^(73, 74). These characteristics explain why partial polarization states in AlScN retain their stability across a wide range of operation speeds^(73 - 76).

From a practical standpoint, the independence of retention from frequency up to 1 MHz is highly encouraging for non-volatile memory applications. It implies that AlScN can support fast access times without sacrificing information stability, a critical requirement for embedded systems and compute-in-memory architecture. Although our present setup could not extend measurements beyond 1 MHz, the robustness of retention at 1 MHz already underscores the suitability of AlScN for high-speed applications.

In summary, our measurements confirm that partial polarization in AlScN enables stable and reliable retention across a broad range of frequencies, highlighting its potential for high-speed nonvolatile

operation. We believe this provides important experimental evidence supporting the practical relevance of partial polarization switching in nitride ferroelectrics.

Change to the manuscript:

The relevant discussion is provided in the **revised Manuscript (pages 10, lines 25 to 32)**. **Figure R21(c)** is added as **Figure 2c** in the **revised Manuscript**. Moreover, in **revised Supplementary Information**, **Figure R21** was added as **Supplementary Figure S12**, presenting the retention characteristics of partial switching in the AlScN capacitor. The results demonstrate that retention in AlScN is unaffected by frequency variation and that partially switched polarization states remain highly stable.

Reviewer #4 (Remarks to the Author):

In this paper, the authors measured the endurance of AlScN capacitors and claimed to achieve record-high performance. While the measurement and analysis are comprehensive, the paper lacks novelty. The data quality is poor, and the analysis is not sufficiently rigorous. The reported improvement in endurance over previous literature is marginal and does not justify publication in Nature Communications. The following are my questions and comments.

Response:

We thank the reviewer for the careful evaluation and for recognizing the broad comprehensive scope of our measurements and analysis. The reviewer's main concerns focus on novelty, data quality, and the clarity of interpretation. We acknowledge these points. In the revised manuscript, we improved the presentation of the results and clarified the structure of the analysis. Our revision was motivated by these issues, and the responses provided here aim to address them. We hope the clarifications and revisions will help resolve the reviewer's concerns.

Reviewer's comments:

1. In Fig. 1d, the leakage current under positive bias is significantly larger than under negative bias. Please explain the underlying physical mechanism responsible for this asymmetry.

Response:

We sincerely thank the reviewer for this important comment. The observation that leakage under positive bias is larger than under negative bias has been repeatedly reported in prior AlScN studies^(25, 77-79), and we observe the same consistent trend across our devices. In our revised manuscript, we describe this behavior as polarization-dependent leakage (PDL). This terminology is also consistent with previous reports in the literature^(25, 77, 78).

PDL is present whether the interface is metal/ferroelectric/metal or metal/ferroelectric/insulator/metal. This holds true regardless of whether the insulator, such as an oxide layer, is located at the top or at the bottom contact. It is consistently observed across different measurement geometries where a symmetric top-electrode drive/sense is used with the bottom electrode

floated⁽⁷⁷⁾. These results confirm that the PDL is not a measurement artifact and cannot be ascribed solely to the location of traps. Prior studies also suggested that this effect cannot be explained only by Schottky barrier lowering at the top or bottom contact ⁽²⁵⁾.

We suggest that this behavior is likely related to the fact that the as-grown film is N-polar, with its polarity established during sputtered deposition. We may also consider that a preferential N-polar state is partly imprinted⁽⁴²⁾ or the preferred polarity. When a positive external bias is applied, this polarity-related internal field acts constructively with the applied field. This lowers the effective barrier at one interface and enhances field-assisted transport, as described in the previous reports^(7, 31). Conversely, when a negative bias is applied, the internal field counteracts the external field, leading to a higher effective barrier and suppressed carrier injection.

Figure R22 | J–V hysteresis of AlScN capacitor with 40 μm diameter at 10 kHz. (b) shows the reorganized J–V curve of the 40 μm device originally presented in **Figure 1d of the main paper. (a) and (c) present magnified plots around the respective switching current peaks.**

However, the PDL cannot be explained solely by a model where current is controlled only by the metal/AlScN interface barrier but is well described by Space-charge-limited current (SCLC). The DC IV behavior, in **Supplementary Figure S1**, also exhibits features consistent with a ferroelectric current response influenced by SCLC. As illustrated in prior SCLC literature⁽³⁷⁾, the hysteresis sequence under $-V_{max} \rightarrow 0 \rightarrow +V_{max} \rightarrow 0 \rightarrow -V_{max}$ follows HRS \rightarrow HRS \rightarrow LRS \rightarrow LRS \rightarrow HRS. In general, ferroelectric DC-IV characteristics exhibit a large hysteresis, with counterclockwise loops under positive bias and clockwise loops under negative bias. However, in **Supplementary Figure S1**, the gap under negative bias is notably reduced compared to the positive side. This asymmetry points to a PDL consistent with SCLC-governed transport. Furthermore, the current slope around -7 V shows a distinct moderation in its magnitude, where the absolute slope decreases rather than increases. Such a feature is

an expected characteristic under SCLC conduction. These observations reinforce that the leakage asymmetry is not solely determined by interface barrier effects but is well explained within the SCLC framework. Furthermore, the J–V hysteresis in **Figure 1d** of the main paper provides clear evidence of PDL governed by SCLC. **Figure R22**, which magnifies the relevant portions of **Figure 1d**, illustrates clockwise current loops under negative bias, indicating increased resistance after $2P_T$ switching. In contrast, under positive bias the current rises after $2P_T$ switching, corresponding to reduced resistance. This polarity-dependent behavior demonstrates that the response remains governed by SCLC-driven PDL.

Within the SCLC framework, our carrier injection is interpreted as a complementary process. The internal field of polarity preference and a graded trap distribution of SCLC set how carriers are injected. Under positive bias, the N-polar imprint adds to the external field and helps the film enter a trap-filled regime at a lower voltage, so current rises. Under negative bias, the internal field counteracts carrier injection to trap. As a result, the film stays in a trap-limited regime, and the current is suppressed. These effects consistently explain the higher current under positive bias and the observed SCLC-driven PDL.

We acknowledge that the microscopic mechanism remains to be fully established. In future work, we plan to conduct temperature-dependent I–V analysis, barrier extraction under both polarities, and further probe measurements (e.g., KPFM and C-AFM) to separate barrier-lowering and trap-assisted contributions.

In summary, the larger leakage under positive bias is a phenomenon consistent with previous reports. It is most reasonably explained by a polarity-imprinted internal field combined with SCLC effects. However, we acknowledge that there is not yet direct experimental evidence to fully support this interpretation, and further investigation is required.

Change to the manuscript:

In the **revised manuscript**, we added PDL and SCLC descriptions to clarify the higher leakage for positive applied voltages (**pages 3, lines 29–32**). These results confirm that the asymmetry originates from polarity-assisted internal fields combined with SCLC transport.

Reviewer’s comments:

2. The authors use P/area (N/area) values instead of P–U/area (N–D/area) to estimate the partial

polarization. This method can lead to an overestimation of remanent polarization, as the charge from leakage current is also included. Please correct the corresponding remanent polarization values to allow a fair comparison with the literature.

Response:

We thank the reviewer for this helpful suggestion. We have revised the **Figure 3** as **Figure R23** and now report both P/Area and N/Area as well as P-U/Area and N-D/Area to avoid confusion. In our study, the values referred to as conventional $2P_r$ were consistently obtained using P-U/Area and N-D/Area. The use of P/Area and N/Area to represent partial polarization appeared only in **Figure 3** of the main paper.

Our use of P/Area and N/Area was motivated by the intrinsic limitation of the PUND method. Since PUND assumes complete polarization switching, it is not ideally suited for quantifying partial polarization states. In partial switching, the U and D pulses can still include switching current, leading to an underestimation of the actual switched polarization if only P-U/Area and N-D/Area are considered. For this reason, we labeled P/Area and N/Area as upper bounds for partial polarization, as explained in **revised Supplementary Information S4** and illustrated in **revised Supplementary Figure S6**.

Figure R23 (c) shows that the conventional $2P_r$ under positive bias yields a negative value for $2P_r$ up to 10^6 cycles, making it invisible on a logarithmic scale. This arises from SCLC-PDL in positive electric fields. Thus, the negative $2P_r$ error in PUND test on AlScN, caused by SCLC-driven PDL, is not only observed in our results but has also been consistently reported in prior studies.^(78, 80 - 82)

Figure R23 | Endurance with Upper bound and Conventional $2P_r$ (c, d) show endurance at preset

conventional $2P_r$ of $10 \mu\text{C}/\text{cm}^2$ and $200 \mu\text{C}/\text{cm}^2$. The black lines represent the upper bound $2P_r$ defined as P/Area for the positive panel and N/Area for the negative panel. The green lines represent the conventional $2P_r$ defined as $(P-U)/\text{Area}$ for the positive panel and $(N-D)/\text{Area}$ for the negative Panel. These two figures are reorganized figures from **Figure 3** in our main paper. **(c)** The upper bound $2P_r$ stays near $34\text{--}38 \mu\text{C}/\text{cm}^2$. **(d)** It lies around $240\text{--}308 \mu\text{C}/\text{cm}^2$ while the conventional $2P_r$ is held at $200 \mu\text{C}/\text{cm}^2$. The red curves show the applied voltage adjusted to maintain the preset $2P_r$.

Particularly, at low voltage, corresponding to the partial polarization regime shown in **Figure R23 (c)**, the switching speed of dipoles in the first pulse P is slow. This is well described in **revised Supplementary information S2**. During the P pulse, switching progresses slowly, leading to a gradual resistance change governed by PDL. The resistance remains high at the beginning but decreases toward the end of the pulse, indicating that the resistance continuously evolves during the P pulse. By contrast, in the subsequent U pulse the resistance is already fully lowered, so the PDL contribution appears more strongly in U . Therefore, the U current exceeds the P current. As a result, the $P - U$ becomes negative. This is a simple consequence of slow switching kinetics and PDL.

However, as the voltage increases, the switching speed becomes faster and the current peak during the P pulse appears earlier, as described in **revised Supplementary Information S2**. This is also consistent with the Kolmogorov–Avrami–Ishibashi (KAI) and nucleation-limited switching (NLS) descriptions of switching kinetics^(34, 35, 83). Here, the high voltage case corresponds to **Figure R23 (d)**, which represents the full switching regime. Therefore, at high voltage the switching during the P pulse is very fast. The device already enters a low resistance state at the beginning of the P pulse. The leakage in P and U is then comparable and large. The conventional $2P_r$ can be obtained as a positive value.

With cycling, the absolute values of positive and negative conventional $2P_r$ gradually become closer in magnitude. We suggest that this trend indicates a progressive reduction of the leakage difference from the PDL. One possible origin is the relaxation or redistribution of traps associated with the as-grown N -polar imprint, which weakens SCLC and thereby reduces injection asymmetry under positive fields⁽³⁷⁾. Direct experimental confirmation will require future targeted studies.

In summary, we now report both metrics side by side and define P/Area and N/Area as upper bounds for partial polarization. The occurrence of negative conventional $2P_r$ at low voltage, followed by recovery at higher voltage and after cycling, is consistent with PDL combined with established

switching kinetics. Similar behavior has been reported in related AIscN studies.

Change to the manuscript:

In the **revised Manuscript**, a concise explanation is provided on **page 14, lines 1–3** and **page 12, line 6**. Extended discussion with PDL and SCLC mechanisms is included at the end of **Supplementary Information S2 (page 32, lines 8-15)**.

Reviewer’s comments:

3. In Fig. 2a, the third phase of the fixed-voltage test shows an increase in the conventional $2P_r$, which the authors attribute to leakage current. However, in principle, the PUND method should subtract the leakage contribution using two consecutive pulses. Please clarify how leakage current could lead to an apparent increase in the extracted $2P_r$.

Response:

We thank the reviewer for the thoughtful question and for pointing out the apparent increase in the conventional $2P_r$ during the third phase of the fixed voltage test. Below we clarify how leakage-related transients can still yield an apparent increase in the conventional $2P_r$ even when a PUND measurement is used.

During endurance measurements, the N pulse exhibits a long current relaxation tail that extends into the subsequent D pulse. As a result, the current in D exhibits lower leakage compared to N, as shown in **revised Supplementary Information S5**. Consequently, when integrating charge over each pulse window we obtain $N = N_{\text{switch}} + N_{\text{leakage}}$ and $D = D_{\text{leakage}}$ with $N_{\text{leakage}} > D_{\text{leakage}}$. Hence, the conventional $2P_r$ (based on P–U and N–D differences) can be overestimated. Here, the long current relaxation tail refers to a decay response that continues beyond the ideal RC delay. This extended relaxation adds a non-switching component, which artificially increases the conventional $2P_r$. This behavior can be explained by defect trap and detrapping mechanisms.

A more fundamental explanation is as follows. Repeated cycling generates defects that enable carriers to be trapped or detrapped. During the P and U pulses, electrons from the top electrode are injected and begin to occupy trap sites. When the N pulse is applied, electrons rapidly fill traps near the bottom electrode, while previously trapped electrons near the top electrode detrapping more slowly^(84 - 86). Because

detrapping is slower than the intrinsic ferroelectric switching speed, it produces a gradual relaxation current that sustains the long current tail. This constitutes one origin of the non-switching contributions. By the time the D pulse is applied, much of the detrapping has already occurred during N pulse, reducing the non-switching contribution and lowering the current. As a result, N and D no longer share the same leakage background. This asymmetry becomes increasingly clear as the trap density increases under endurance test electrical stress, leading to an overestimation of conventional $2P_r$. A similar process occurs under positive voltage. When a P pulse is applied, electrons become trapped at the top electrode. At the same time, electrons at the bottom electrode that were trapped during the prior N and D pulses detrapp gradually. This delayed detrapping contributes to the overall relaxation process. This behavior is consistently observed regardless of our voltage adjustment algorithm. The PUND current responses shown in **Supplementary Figures S17 and S18** and **Supplementary Information S5** provide direct evidence of this behavior. In these results, P and N show higher leakage currents, while U and D display lower leakage. Consequently, after extended cycling, both positive and negative branches can be overestimated due to defect-mediated effects.

These mechanisms can be explained by Maxwell Wagner relaxation and Curie von Schweidler relaxation. In Maxwell Wagner relaxation, conductivity and permittivity mismatch, such as at the interface between the Al electrode and the AlScN film, the interface causes charge to accumulate during the pulse⁽⁸⁾. During the pulse, traps fill quickly under the applied field. After the pulse, charge releases slowly because detrapping is limited by interface traps with a broad distribution of time constants. In the trap picture, traps fill quickly during the pulse, whereas detrapping after the pulse is slow through interface traps with broad time constants. As a result, this process produces the Curie von Schweidler relaxation current. This relaxation current accounts for the long current tail observed in our measurements. In addition, high field cycling increases the interfacial layer thickness and accelerates these relaxation effects⁽⁴²⁾. This produces fatigue, slows relaxation currents, and strengthens the asymmetry between N and D. As a result, the long current relaxation tail extends beyond an ideal RC delay and the conventional $2P_r$ is overestimated.

In addition to the major reasons described above, several other factors can further intensify the overestimation. PDL in AlScN MFM capacitors has been repeatedly reported, which directly leads to N_{leakage} being greater than D_{leakage} and artificially elevates the conventional $2P_r$ in the case of partially polarized AlScN.

Change to the manuscript:

In the **revised manuscript**, we clarified that the apparent increase in conventional $2P_r$ during fixed-voltage cycling originates from long relaxation tails of the N pulse current extending into the D pulse (**page 9, lines 10-13**). Further discussion is provided at the end of revised **Supplementary information S5 (page 50, lines 17 - 23)**. This causes N to include additional leakage contributions compared to D, leading to $N > D$ and an overestimation of conventional $2P_r$. We attribute this behavior to defect-mediated trap and detrapping processes that produce slow relaxation currents beyond the ideal RC response.

Reviewer's comments:

4. In the endurance test, as the device approaches its end-of-life, the leakage current increases significantly. In this regime, the leakage current is no longer negligible during polarization measurements, resulting in substantial errors in the “intrinsic $2P_r$ ” extraction method. Consequently, the endurance number obtained using this method is not reliable.

Response:

We sincerely thank the reviewer for pointing out this important aspect. We agree that, as the device approaches its end-of-life, the rapid increase in leakage current makes the extraction of the so-called “intrinsic $2P_r$ ” unreliable. PUND measurement itself is the most widely adopted and efficient method for isolating the ferroelectric switching response, but it still suffers from limitations, where leakage and non-switching contributions cannot be fully separated.

Nevertheless, during endurance cycling we continuously observed reproducible explicit ferroelectric switching peaks. The devices also survived and show these peaks until very high cycle counts, as shown in **revised Supplementary Information S5 and revised Supplementary Figures S17 and S18**. This confirms that the endurance numbers themselves are valid, although the exact fraction of switched polarization cannot be unambiguously quantified in this regime. We emphasize that this limitation is methodological rather than a failure of real device operation.

Change to the manuscript:

In the **revised Supplementary Information**, we further added that reproducible switching peaks confirm survival to high cycle counts (**pages 50, lines 17-23**).

Reviewer's comments:

5. For Fig. 4, please include plots of both the “upper bound 2Pr” and the “lower bound 2Pr” (conventional 2Pr) to enable a fair comparison with other works.

Response:

We sincerely thank the reviewer for this constructive suggestion. In **Figure 4** of our main paper, we originally presented only the “lower bound 2Pr” (conventional 2Pr, obtained from (P–U)/Area and (N–D)/Area pulses) to maintain consistency with prior reports and to avoid confusion in benchmarking. However, we agree with the reviewer that including both “upper bound 2Pr” (P/Area and N/Area) and “lower bound 2Pr” allows for a more comprehensive comparison across different studies. The inclusion of upper bound 2Pr is particularly useful in the partial switching regime. In this regime, conventional PUND analysis underestimates the switched polarization, which makes the performance appear lower than it is. Following the reviewer's advice, we have revised **Figure 4** to report both upper and lower bound 2Pr values, thereby offering a clearer benchmark for endurance and enabling a fairer comparison with prior studies.

Figure R24 | Benchmarking of ferroelectric endurance performance. A systematic comparison of AlScN endurance characteristics with previously reported ferroelectric materials. The red markers represent data from this study. Benchmarking maximum endurance cycles against effective 2Pr. This figure is replaced as **Figure 4a** and **c** in our main paper.

Change to the manuscript:

In the revised manuscript, **Figure R24** is reorganized in **Figure 4**.

Reference

1. S. Fichtner, N. Wolff, F. Lofink, L. Kienle, B. Wagner, AlScN: A III-V semiconductor based ferroelectric, *J. Appl. Phys.*, **125**, 114103 (2019)
2. M. Akiyama, T. Kamohara, K. Kano, A. Teshigahara, Y. Takeuchi, N. Kawahara, Enhancement of Piezoelectric Response in Scandium Aluminum Nitride Alloy Thin Films Prepared by Dual Reactive Cosputtering, *Adv. Mater.*, **21**, 593–596 (2009)
3. M. Lanza, H.-S. P. Wong, E. Pop, D. Ielmini, D. Strukov, B. C. Regan, L. Larcher, M. A. Villena, J. J. Yang, L. Goux, A. Belmonte, Y. Yang, F. M. Puglisi, J. Kang, B. Magyari-Köpe, E. Yalon, A. Kenyon, M. Buckwell, A. Mehonic, A. Shluger, H. Li, T.-H. Hou, B. Hudec, D. Akinwande, R. Ge, S. Ambrogio, J. B. Roldan, E. Miranda, J. Suñe, K. L. Pey, X. Wu, N. Raghavan, E. Wu, W. D. Lu, G. Navarro, W. Zhang, H. Wu, R. Li, A. Holleitner, U. Wurstbauer, M. C. Lemme, M. Liu, S. Long, Q. Liu, H. Lv, A. Padovani, P. Pavan, I. Valov, X. Jing, T. Han, K. Zhu, S. Chen, F. Hui, Y. Shi, Recommended Methods to Study Resistive Switching Devices, *Adv. Electron. Mater.*, **5**, 1800143 (2019)
4. Y. Zuo, H. Lin, J. Guo, Y. Yuan, H. He, Y. Li, Y. Xiao, X. Li, K. Zhu, T. Wang, X. Jing, C. Wen, M. Lanza, Effect of the Pressure Exerted by Probe Station Tips in the Electrical Characteristics of Memristors. *Adv. Electron. Mater.*, **6**, 1901226 (2020)
5. K. N. Tu, Yingxia Liu, Menglu Li, Effect of Joule heating and current crowding on electromigration in mobile technology. *Appl. Phys. Rev.*, **4** (1): 011101 (2017)
6. C.W. Lee, K. Yazawa, A. Zakutayev, G.L. Brennecka, P. Gorai, Switching it up: new mechanisms revealed in wurtzite-type ferroelectrics, *Sci. Adv.*, **10** (20) (2024)
7. KH. Kim, Z. Han, Y. Zhang, P. Musavigharavi, J. Zheng, D. K. Pradhan, E. A. Stach, R. H. Olsson III, D. Jariwala, *ACS Nano* **18** (24), 15925-15934 (2024)
8. J Liu, CG. Duan, WG. Yin, W. N. Mei, R. W. Smith, J. R. Hardy, Large dielectric constant and Maxwell-Wagner relaxation in $\text{Bi}_{2/3}\text{Cu}_3\text{Ti}_4\text{O}_{12}$, *Phys. Rev. B* **70**, 144106 (2004)
9. R. Bouregba, On the origin of polarization fatigue and Curie–von Schweidler relaxation current in $\text{Pb}(\text{Zr}_x\text{Ti}_{1-x})\text{O}_3$ ferroelectric thin films: A unique mechanism based on charge trapping by interface defects, *J. Appl. Phys.* **133**, 014101 (2023)
10. X. Cheng, C. Zhou, B. Lin, Z Yang, S Chen, K. H.L. Zhang, Z. Chen, Leakage mechanism in ferroelectric $\text{Hf}_{0.5}\text{Zr}_{0.5}\text{O}_2$ epitaxial thin films, *Applied Materials Today*, **32**, 101804 (2023)
11. S. S. Fields, S. W. Smith, S. T. Jaszewski, T. Mimura, D. A. Dickie, G. Esteves, M. D. Henry, S. L. Wolfley, P. S. Davids, J. F. Ihlefeld, Wake-up and fatigue mechanisms in

- ferroelectric $\text{Hf}_{0.5}\text{Zr}_{0.5}\text{O}_2$ films with symmetric RuO_2 electrodes. *J. Appl. Phys.*, **130** (13), 134101 (2021)
12. E. D. Grimley, T. Schenk, X. Sang, M. Pešić, U. Schroeder, T. Mikolajick, J. M. LeBeau, Structural Changes Underlying Field-Cycling Phenomena in Ferroelectric HfO_2 Thin Films. *Adv. Electron. Mater.*, **2**, 1600173 (2016)
 13. Y. Cheng, Z. Gao, K. H. Ye, H. W. Park, Y. Zheng, Y. Zheng, J. Gao, M. H. Park, JH Choi, KH Xue, C. S. Hwang, H. Lyu, Reversible transition between the polar and antipolar phases and its implications for wake-up and fatigue in HfO_2 -based ferroelectric thin film. *Nat Commun* **13**, 645 (2022).
 14. S. Li, D. Zhou, Z. Shi, M. Hoffmann, T. Mikolajick, U. Schroeder, Involvement of Unsaturated Switching in the Endurance Cycling of Si-doped HfO_2 Ferroelectric Thin Films. *Adv. Electron. Mater.*, **6**, 2000264 (2020)
 15. C. Wang, H. Qiao, Y. Kim, Perspective on the switching behavior of HfO_2 -based ferroelectrics. *J. Appl. Phys.*, **129** (1), 010902 (2021)
 16. X. Li, P. Srivari, E. Paasio, S. Majumda, Understanding fatigue and recovery mechanisms in $\text{Hf}_{0.5}\text{Zr}_{0.5}\text{O}_2$ capacitors for designing high endurance ferroelectric memory and neuromorphic hardware, *Nanoscale*, **17**, 6058-6071 (2025)
 17. A. Mallick, M. K. Lenox, T. E. Beechem, J. F. Ihlefeld, N. Shukla, Oxygen vacancy contributions to the electrical stress response and endurance of ferroelectric hafnium zirconium oxide thin films. *Appl. Phys. Lett.*, **122** (13), 132902 (2023)
 18. K. Yazawa, C. Evans, E. C. Dickey, M. B. Tellekamp, G. L. Brennecka, A. Zakutayev, Low leakage current in heteroepitaxial $\text{Al}_{0.7}\text{Sc}_{0.3}\text{N}$ ferroelectric films on GaN, *Phys. Rev. Applied* **23**, 014036 (2025)
 19. J. X. Zheng, M. M. A. Fiagbenu, G. Esteves, P. Musavigharavi, A. Gunda, D. Jariwala, E. A. Stach, R. H. Olsson, Ferroelectric behavior of sputter deposited $\text{Al}_{0.72}\text{Sc}_{0.28}\text{N}$ approaching 5 nm thickness. *Appl. Phys. Lett.*, **122** (22), 222901 (2023)
 20. J. Su, S. Fichtner, M. Z. Ghori, N. Wolff, M. R. Islam, A. Lotnyk, D. Kaden, F. Niekkel, L. Kienle, B. Wagner, F. Lofink, Growth of Highly c-Axis Oriented AlScN Films on Commercial Substrates, *Micromachines (Basel)*, **13**(5), 783 (2022)
 21. V. V. Felmetger, M. K. Mikhov, Deposition of smooth and highly (111) textured Al bottom electrodes for AlN-based electroacoustic devices, *2012 IEEE International Frequency Control Symposium Proceedings*, Baltimore, MD, USA, pp. 1-4 (2012)

22. S. Wang, V. Dhyani, S. S. Mohanraj, X. Shi, B. Varghese, W. W. Chung, D. Huang, Z. Shiuh Lim, Q. Zeng, H. Liu, X. Luo, V. Leong, N. Li, D. Zhu, CMOS-compatible photonic integrated circuits on thin-film ScAlN. *APL Photonics*, 9 (6), 066109 (2024)
23. R. Nie, S. Shao, Z. Luo, X. Kang, T. Wu, Characterization of Ferroelectric Al_{0.7}Sc_{0.3}N Thin Film on Pt and Mo Electrodes, *Micromachines* 13, no. 10, 1629 (2022)
24. Y. Lu, M. Reusch, N. Kurz, A. Ding, T. Christoph, M. Prescher, L. Kirste, O. Ambacher, A. Žukauskaitė, Elastic modulus and coefficient of thermal expansion of piezoelectric Al_{1-x}Sc_xN (up to x = 0.41) thin films. *APL Mater.*, 6 (7), 076105 (2018)
25. D. Wang, P. Musavigharavi, J. Zheng, G. Esteves, X. Liu, M. M. A. Fiagbenu, E. A. Stach, D. Jariwala, R. H. Olsson III, Sub-Microsecond Polarization Switching in (Al,Sc)N Ferroelectric Capacitors Grown on Complementary Metal–Oxide–Semiconductor-Compatible Aluminum Electrodes. *Phys. Status Solidi RRL*, 15: 2000575 (2021)
26. S. Barth, T. Schreiber, S. Cornelius, O. Zywitzki, T. Modes, H. Bartzsch, High Rate Deposition of Piezoelectric AlScN Films by Reactive Magnetron Sputtering from AlSc Alloy Targets on Large Area, *Micromachines (Basel)*, 13(10), 1561 (2022).
27. Z. Tang, G. Esteves, R. H. Olsson III, Sub-quarter micrometer periodically poled Al_{0.68}Sc_{0.32}N for ultra-wideband photonics and acoustic devices, *J. Appl. Phys.* 134, 114101 (2023)
28. H. Lu, G. Schönweger, A. Petraru, H. Kohlstedt, S. Fichtner, A. Gruverman, Domain Dynamics and Resistive Switching in Ferroelectric Al_{1-x}Sc_xN Thin Film Capacitors, *Adv. Funct. Mater.*, 34, 2315169 (2024)
29. H. Lu, G. Schönweger, N. Wolff, Z. Ding, A. Petraru, I. Streicher, H. Kohlstedt, C. Kübel, S. Leone, L. Kienle, S. Fichtner, A. Gruverman, Al_{1-x}Sc_xN-Based Ferroelectric Domain-Wall Memristors. *Adv. Funct. Mater.*, 2503143 (2025)
30. D. Drury, K. Yazawa, A. Zakutayev, B. Hanrahan, G. Brennecka, High-Temperature Ferroelectric Behavior of Al_{0.7}Sc_{0.3}N, *Micromachines (Basel)*. May 31;13(6), 887 (2022)
31. X. Liu, J. Zheng, D. Wang, P. Musavigharavi, E. A. S., R. Olsson, D. Jariwala, Aluminum scandium nitride-based metal–ferroelectric–metal diode memory devices with high on/off ratios. *Appl. Phys. Lett.*, 118 (20), 202901 (2021)
32. P. Wang, D. Wang, S. Mondal, Z. Mi, Ferroelectric N-polar ScAlN/GaN heterostructures grown by molecular beam epitaxy. *Appl. Phys. Lett.*, 121 (2), 023501 (2022)
33. Z. Hu, H. Cho, R. K. Rai, K. Bao, Y. Zhang, Z. Qu, Y. He, Y. Ji, C. Leblanc, KH Kim, Z. Han, Z. Qiu, X. Du, E. A. Stach, R. Olsson, D. Jariwala, Demonstration of Highly Scaled AlScN

- Ferroelectric Diode Memory with a Storage Density of >100 Mbit/mm², *Nano Letters* 25 (37), 13748-13755 (2025)
34. Y. Yang, T. Zhang, Z. Zheng, et al. Thickness-Driven Transition of Switching Kinetics in Wurtzite Ferroelectrics. *Adv. Funct. Mater.* (2025)
 35. Z. Chen, Y. Zhang, S. Li, XM. Lu, W. Cao, Frequency dependence of the coercive field of 0.71Pb(Mg_{1/3}Nb_{2/3})O₃-0.29PbTiO₃ single crystal from 0.01 Hz to 5 MHz. *Appl Phys Lett.*, 110(20):202904 (2017)
 36. G. Schönweger, M. R. Islam, N. Wolff, A. Petraru, L. Kienle, H. Kohlstedt, S. Fichtner, Ultrathin Al_{1-x}Sc_xN for Low-Voltage-Driven Ferroelectric-Based Devices. *Phys. Status Solidi RRL*, 17: 2200312 (2023)
 37. J. Tian, Z. Tan, Z. Fan, D. Zheng, Y. Wang, Z. Chen, F. Sun, D. Chen, M. Qin, M. Zeng, X. Lu, X. Gao, JM. Liu, Depolarization-Field-Induced Retention Loss in Ferroelectric Diodes, *Phys. Rev. Applied* 11, 024058 (2019)
 38. F. P. G. Fengler, M. Hoffmann, S. Slesazeck, T. Mikolajick, U. Schroeder, On the relationship between field cycling and imprint in ferroelectric Hf_{0.5}Zr_{0.5}O₂. *J. Appl. Phys.*, 123 (20) 204101 (2018)
 39. M. Saadi , W. Shao , M. Zhang, H. Qin, C. Han, Y. Hu, H. Zhang, X. Wang, Y. Tong, Exploring the underlying mechanisms of ferroelectric behavior in metal-doped aluminum nitride: an in-depth review. *Microstructures*, 5, 2025092 (2025)
 40. K. Yazawa, D. Drury, J. Hayden, JP. Maria, S. Trolier-McKinstry, A. Zakutayev, G. L. Brenneka, Polarity effects on wake-up behavior of Al_{0.94}B_{0.06}N ferroelectrics. *J Am Ceram Soc.* 107: 1523–1532 (2024)
 41. SH. Teng, A. Dimou, B. Udofia, M. Ghasemi, M. Stricker, A. Grünebohm, Control of ferroelectric domain wall dynamics by point defects: Insights from *ab initio* based simulations. *J. Appl. Phys.* 137 (15): 154103 (2025)
 42. K. D. Kim, Y. B. Lee, S. H. Lee, I. S. Lee, S. K. Ryoo, S. Byun, J. H. Lee, H. Kim, H. W. Park, C. S. Hwang, Evolution of the Ferroelectric Properties of AlScN Film by Electrical Cycling with an Inhomogeneous Field Distribution. *Adv. Electron. Mater.*, 0, 2201142 (2023)
 43. V. Gund, B. Davaji, H. Lee, M. J. Asadi, J. Casamento, H. G. Xing, D. Jena, A. Lal, Temperature-dependent Lowering of Coercive Field in 300 nm Sputtered Ferroelectric Al_{0.70}Sc_{0.30}N, 2021 IEEE International Symposium on Applications of Ferroelectrics (ISAF), Sydney, Australia, pp. 1-3 (2021)

44. T. Schenk, M. Hoffmann, J. Ocker, M. Pešić, T. Mikolajick, U. Schroeder, *ACS Applied Materials & Interfaces*, 7 (36), 20224-20233 (2015)
45. K. D. Kim, Y. B. Lee, S. H. Lee, I. S. Lee, S. K. Ryoo, S. Y. Byun, J. H. Lee, C. S. Hwang, Impact of operation voltage and NH₃ annealing on the fatigue characteristics of ferroelectric AlScN thin films grown by sputtering, *Nanoscale*, 15, 16390-16402 (2023)
46. S.L. Tsai, T. Hoshii, H. Wakabayashi, K. Tsutsui, TK. Chung, E. Y. Chang, K. Kakushima, Field cycling behavior and breakdown mechanism of ferroelectric Al_{0.78}Sc_{0.22}N films, *Jpn. J. Appl. Phys.* **61** SJ1005 (2022)
47. R. Guido, T. Mikolajick, U. Schroeder, P. D. Lomenzo, Role of Defects in the Breakdown Phenomenon of Al_{1-x}Sc_xN: From Ferroelectric to Filamentary Resistive Switching, *Nano Lett*, 23, 15, 7213–7220 (2023)
48. S. Fichtner, G. Schönweger, CW. Lee, K. Yazawa, P. Gorai, G. L. Brennecka, Polarization and domains in wurtzite ferroelectrics: Fundamentals and applications. *Appl. Phys. Rev.*, 12 (2): 021310 (2025)
49. S. Calderon V, J. Hayden, S. M. Baksa, W. Tzou, S. Trolier-McKinstry, I. Dabo, JP. Maria, Elizabeth C. Dickey, Atomic-scale polarization switching in wurtzite ferroelectrics, *Science*, **380**, 1034-1038 (2023)
50. A. K. Tagantsev, I. Stolichnov, N. Setter, J. S. Cross, M. Tsukada, Non-Kolmogorov-Avrami switching kinetics in ferroelectric thin films, *Phys. Rev. B*, **66**, 214109 (2002)
51. Y. Kim, HH. Han, W. Lee, S. Baik, D. Hesse, M. Alexe, Non-Kolmogorov–Avrami–Ishibashi Switching Dynamics in Nanoscale Ferroelectric Capacitors *Nano Letters* 10 (4), 1266-1270 (2010)
52. Y. Ahn, J. Y. Son, Activation field-driven domain wall dynamics of nanobits in ferroelectric Al_{0.7}Sc_{0.3}N thin films, *Journal of Alloys and Compounds*, 1035, 5, 181529 (2025)
53. R. Guido, H. Lu, P. D Lomenzo, T. Mikolajick, A. Gruverman, U. Schroeder, Kinetics of N-to M-Polar Switching in Ferroelectric Al_{1-x}Sc_xN Capacitors. *Adv Sci (Weinh)*. Apr, 11(16):e2308797 (2024).
54. R. Bulanadi, K. Cordero-Edwards, P. Tückmantel, S. Saremi, G. Morpurgo, Q. Zhang, L. W. Martin, V. Nagarajan, P. Paruch, Interplay between Point and Extended Defects and Their Effects on Jerky Domain-Wall Motion in Ferroelectric Thin Films, *Phys. Rev. Lett.* **133**, 106801 (2024)

55. CW. Lee, N. U. Din, G. L. Brennecka, P. Gorai, Defects and oxygen impurities in ferroelectric wurtzite $\text{Al}_{1-x}\text{Sc}_x\text{N}$ alloys. *Appl. Phys. Lett.*, 125 (2): 022901 (2024)
56. G. Catalan, J. Seidel, R. Ramesh, J. F. Scott, Domain wall nanoelectronics, *Rev. Mod. Phys.* **84**, 119 (2012)
57. Y. Zhan, Q. Zhu, B. Tian, C. Duan, New-Generation Ferroelectric AlScN Materials, *Nanomicro Lett.*, 16:227 (2024)
58. M. Y. Gureev, P. Mokrý, A. K. Tagantsev, N. Setter, Ferroelectric charged domain walls in an applied electric field, *Phys. Rev. B* **86**, 104104 (2012)
59. M. T. Do, N. Gauquelin, M. D. Nguyen, F. Blom, J. Verbeeck, G. Koster, E. P. Houwman, G. Rijnders, Interface degradation and field screening mechanism behind bipolar-cycling fatigue in ferroelectric capacitors. *APL Mater.*, 9 (2): 021113 (2021)
60. D. R. Småbråten, T. S. Holstad, D. M. Evans, Z. Yan, E. Bourret, D. Meier, S. M. Selbach, Domain wall mobility and roughening in doped ferroelectric hexagonal manganites, *Phys. Rev. Research* **2**, 033159 (2020)
61. P. S. Bednyakov, T. Sluka, A. K. Tagantsev, D. Damjanovic, N. Setter, Formation of charged ferroelectric domain walls with controlled periodicity, *Scientific Reports*, volume 5, Article number: 15819 (2015)
62. T. Hwang, W. Aigner, T. Metzger, A. C. Kummel, K. Cho, First-Principles Understanding on the Formation of Inversion Domain Boundaries of Wurtzite AlN, AlScN, and GaN, *ACS Applied Electronic Materials* 6 (5), 3257-3263(2024)
63. S. Song, D. K. Pradhan, Z. Hu, Y. Zhang, R. N. Keneipp, M. A. Susner, P. Bhattacharya, M. Drndić, R. H. Olsson III, D. Jariwala, Observation of giant remnant polarization in ultrathin AlScN at cryogenic temperatures, arXiv:2503.19491 (2025)
64. L. Chen, C. Liu, H. K. Lee, B. Varghese, R. W. F. Ip, M. Li, Z. J. Quek, Y. Hong, W. Wang, W. Song, H. Lin, Y. Zhu, Demonstration of 10 nm Ferroelectric $\text{Al}_{0.7}\text{Sc}_{0.3}\text{N}$ -Based Capacitors for Enabling Selector-Free Memory Array, *Materials*, 17(3), 627 (2024)
65. X. Li, P. Srivari, E. Paasio, S. Majumdar, Understanding fatigue and recovery mechanisms in $\text{Hf}_{0.5}\text{Zr}_{0.5}\text{O}_2$ capacitors for designing high endurance ferroelectric memory and neuromorphic hardware, *Nanoscale*, 17, 6058-6071 (2025)

66. H. Bohuslavskyi, K. Grigoras, M. Ribeiro, M. Prunnila, S. Majumdar, Ferroelectric $\text{Hf}_{0.5}\text{Zr}_{0.5}\text{O}_2$ for Analog Memory and In-Memory Computing Applications Down to Deep Cryogenic Temperatures. *Adv. Electron. Mater.*, 10, 2300879 (2024)
67. Z. Gao, Y. Luo, S. Lyu, Y. Cheng, Y. Zheng, Q. Zhong, Identification of Ferroelectricity in a Capacitor With Ultra-Thin (1.5-nm) $\text{Hf}_{0.5}\text{Zr}_{0.5}\text{O}_2$ Film, *IEEE Electron Device Letters*, vol. 42, no. 9, pp. 1303-1306, (2021)
68. J. Bouaziz, P. R. Romeo, N. Baboux, B. Vilquin, Huge Reduction of the Wake-Up Effect in Ferroelectric HZO Thin Films, *ACS Applied Electronic Materials* 1 (9), 1740-1745 (2019)
69. Y. Cao, W. Zhang Y. Li, Hafnium-doped zirconia ferroelectric thin films with excellent endurance at high polarization, *Nanoscale*, 15, 1392-1401 (2023)
70. J.L. Le, Z. P. Bazant, M. Z. Bazant, Lifetime of high- k gate dielectrics and analogy with strength of quasibrittle structures, *J. Appl. Phys.* 106, 104119 (2009)
71. E. Y. Wu, J. Sune and W. Lai, On the Weibull shape factor of intrinsic breakdown of dielectric films and its accurate experimental determination. Part II: experimental results and the effects of stress conditions, in *IEEE Transactions on Electron Devices*, vol. 49, no. 12, pp. 2141-2150, (2002)
72. J.L. Le, A finite weakest-link model of lifetime distribution of high- k gate dielectrics under unipolar AC voltage stress, *Microelectronics Reliability*, Volume 52, Issue 1, Pages 100-106 (2012)
73. D. Zhao, T. Lenz, G. H. Gelinck, P. Groen, D. Damjanovic, D. M. de Leeuw, I. Katsouras, Depolarization of multidomain ferroelectric materials. *Nat Commun* 10, 2547 (2019)
74. B. Darinskii, A. Sidorkin, A. Sigov, N. Popravko, Influence of Depolarizing Fields and Screening Effects on Phase Transitions in Ferroelectric Composites. *Materials (Basel)*. Jan 6;11(1):85 (2018)
75. K. D. Kim, S. K. Ryoo, H. S. Park, S. Choi, T. W. Park, M. K. Yeom, C. S. Hwang, Comparative study on the stability of ferroelectric polarization of HfZrO_2 and AlScN thin films over the depolarization effect. *J. Appl. Phys.*, 136 (2): 024101 (2024)
76. G. A. Salcedo, M. Harrington, S. Nikodemski, V. Vasilyev, M. Newburger, T. Wolfe, C. S. Kabban, J. Sattler, A. Islam, A statistical study on the origin of the polarization-dependent leakage in ferroelectric aluminum scandium nitride films. *J. Appl. Phys.*, 138 (4): 044104 (2025)

77. D. Wang, J. Zheng, P. Musavigharavi, W. Zhu, A. C. Foucher, S. E. Trolrier-McKinstry, Ferroelectric Switching in Sub-20 nm Aluminum Scandium Nitride Thin Films, in *IEEE Electron Device Letters*, vol. 41, no. 12, pp. 1774-1777, (2020)
78. KH Kim, S. Oh, M. M. A. Fiagbenu, J. Zheng, P. Musavigharavi, P. Kumar, N. Trainor, A. Aljarb, Y. Wan, H. M. Kim, K. Katti, S. Song, G. Kim, Z. Tang, JH Fu, M. Hakami, V. Tung, J. M. Redwing, E. A. Stach, R. H. Olsson III, D. Jariwala, Scalable CMOS back-end-of-line compatible AlScN/two-dimensional channel ferroelectric field-effect transistors, *Nat. Nanotechnol.*, 18, 1044–1050, (2023)
79. D. K. Pradhan, D. C. Moore, G. Kim, Y. He, P. Musavigharavi, KH Kim, N. Sharma, Z. Han, X. Du, V. S. Puli, E. A. Stach, W. J. Kennedy, N. R. Glavin, R. H. Olsson, D. Jariwala, A scalable ferroelectric non-volatile memory operating at 600 °C, *Nat Electron*, 7, 348–355, (2024)
80. X. Liu, D. Wang, KH. Kim, K. Katti, J. Zheng, P. Musavigharavi, J. Miao, E. A. Stach, R. H. Olsson III, D. Jariwala, *Nano Letters*, 21 (9), 3753-3761 (2021)
81. X. Liu, J. Ting, Y. He, M. M. A. Fiagbenu, J. Zheng, D. Wang, J. Frost, P. Musavigharavi, G. Esteves, K. Kisslinger, S. B. Anantharaman, E. A. Stach, R. H. Olsson III, D. Jariwala, Reconfigurable Compute-In-Memory on Field-Programmable Ferroelectric Diodes, *Nano Letters*, 22, 7690-7698, (2022)
82. D. Wang, J. Zheng, Z. Tang, M. D'Agati, P. S. M. Gharavi, X. Liu, Ferroelectric C-Axis Textured Aluminum Scandium Nitride Thin Films of 100 nm Thickness, *2020 Joint Conference of the IEEE International Frequency Control Symposium and International Symposium on Applications of Ferroelectrics (IFCS-ISAF)*, Keystone, CO, USA, pp. 1-4 (2020)
83. T. D. Cornelissen, I. Urbanaviciute, M. Kemerink, Microscopic model for switching kinetics in organic ferroelectrics following the Merz law, *Phys. Rev. B* **101**, 214301 (2020)
84. Y. Song, P. F. Jiang, P. Xu, X. Y. Peng, Q. Q. Wei, Q. Y. Yan, W. Wei, Y. Wang, X. Long, T. C. Gong, Y. Yang, E. V. Ramana, Q. Luo, Fatigue of ferroelectric field effect transistor: mechanisms and optimization strategies. *J. Semicond.*, 46(6), 061302 (2025)
85. J. Li, M. Si, Y. Qu, X. Lyu and P. D. Ye, "Quantitative Characterization of Ferroelectric/Dielectric Interface Traps by Pulse Measurements," in *IEEE Transactions on Electron Devices*, vol. 68, no. 3, pp. 1214-1220, (2021)

86. Y. Qu, J. Li, M. Si, X. Lyu and P. D. Ye, Quantitative Characterization of Interface Traps in Ferroelectric/Dielectric Stack Using Conductance Method, in *IEEE Transactions on Electron Devices*, vol. 67, no. 12, pp. 5315-5321, (2020)
87. X. J. Lou, Polarization fatigue in ferroelectric thin films and related materials, *J. Appl. Phys.* 105, 024101 (2009)

Reviewer's comment 1: "The manuscript can be accepted in the current revised state."

Our response: Thank you for your positive assessment and endorsement of our revised manuscript.

Reviewer's comment 2: "In the revised manuscript, the authors addressed all of my questions with adequate experimental data. I recommend publishing the manuscript in Nature Communications in its current form."

Our response: We greatly appreciate the reviewer's positive recommendation and the time taken to review our experimental data.

Reviewer's comment 3: "The author has replied to all my questions."

Our response: Thank you for acknowledging our responses.

Reviewer's comment 4: "It remains unclear what the novelty of this paper is. The device structure is very simple, and the measurement methodology is conventional. The data quality appears noisy, and the reported endurance value is only marginally better than previously published results. It is therefore difficult to justify why this work merits publication in Nature Communications."

Our response: We thank the reviewer for their comments. We remain convinced of the novelty and utility of our work given that our Fe AlScN samples are sub 50 nm in thickness where high reliability has not been demonstrated. Further the novelty of our partial switching and endurance measurement approach opens a route to understanding and measuring reliability in ferroelectric AlScN